



# A 29-year time series of annual 300-metre resolution plant functional type maps for climate models

Kandice L. Harper[1], Céline Lamarche[1], Andrew Hartley[2], Philippe Peylin[3], Catherine Ottlé[3], Vladislav Bastrikov[3], Rodrigo San Martín[3], Sylvia I. Bohnenstengel[4], Grit Kirches[5], Martin Boettcher[5], Roman Shevchuk[5], Carsten Brockmann[5], Pierre Defourny[1]

[1]Earth and Life Institute, Université catholique de Louvain, 1348 Louvain-la-Neuve, Belgium
[2]Met Office Hadley Centre, Exeter, EX1 3PB, United Kingdom
[3]Laboratoire des Sciences du Climat et de l'Environnement, Institut Pierre-Simon Laplace, CEA-CNRS-Université Paris-Saclay, Orme des Merisiers, 91191 Gif-sur-Yvette, France
[4]Met Office, Reading, RG6 6BB, United Kingdom
[5]Brockmann Consult GmbH, 21029 Hamburg, Germany

*Correspondence to*: Céline Lamarche (celine.lamarche@uclouvain.be)

**Abstract.** The existing medium-resolution land cover time series produced under the European Space Agency's Climate Change Initiative provides 29 years (1992–2020) of annual land cover maps at 300-m resolution, allowing for a detailed study of land change dynamics over the contemporary era. Because models need 2D parameters rather than 2D land cover information, the land cover classes must be converted into model-appropriate plant functional types (PFTs) to apply this time series to Earth system and land surface models. The first generation cross-walking table that was presented with the land cover product prescribed pixel-level PFT fractional compositions that varied by land cover class but lacked spatial variability. Here we describe a new ready-to-use data product for climate modelling: spatially explicit annual maps of PFT fractional composition at 300 m resolution for 1992–2020, created by fusing the 300 m medium-resolution land cover product with several existing high-resolution datasets using a globally consistent method. In the resulting data product, which has 14 layers for each of the 29 years, pixel values at 300-m resolution indicate the percentage cover (0–100 %) for each of 14 PFTs, with pixel-level PFT composition exhibiting significant intra-class spatial variability at the global scale. We additionally present an updated version of the user tool that allows users to modify the baseline product (e.g., re-mapping, re-projection, PFT conversion, and spatial sub-setting) to meet individual needs. Finally, these new PFT maps have been used in two land surface models - ORCHIDEE and JULES - to demonstrate their benefit over the conventional maps based on a generic cross-walking table. Regional changes in the fractions of trees, short vegetation, and bare soil cover induce changes in surface properties, such as the albedo, leading to significant changes in surface turbulent fluxes, temperature, and vegetation carbon stocks.

## Introduction

Terrestrial ecosystems have always been shaped by people who depend on land for their consumption of direct (e.g., food and materials) and indirect (e.g., land for human activities) goods (Vitousek et al., 1986; Foley et al., 2005). Land cover change induces significant biogeochemical and biogeophysical effects on the climate by altering greenhouse gas emissions (e.g., $CO_2$) and the surface energy budget, induced by modified albedo, evapotranspiration, and roughness (Pielke et al., 2011; Mahmood et al., 2014; Pielke, 2005; Brovkin et al., 2006; Dale, 1997; Liu et al., 2017). The fragmented landscapes that result from land cover change also influence surface temperatures, altering clouds and precipitation (Dale, 1997; Perugini et al., 2017; Sampaio et al., 2007). The physical climate changes driven by land cover change can manifest far afield of the surface changes; for example, large areas deforested at the expense of brighter land cover (e.g., cropland expansion) modify albedo (Loarie et al.,



2011; Lambin et al., 2001), with the altered energy balance driving changes in monsoon patterns (Feddema et al., 2005;
Devaraju et al., 2015).

Anthropogenic activities, driven mainly by economic and population growth (Pachauri and Meyer, 2014), have changed the
atmosphere's composition (IPCC, 2022). The land use, land-use change, and forestry sector is estimated to account for net
emissions of $4.1 \pm 2.6$ Gt $CO_2$ yr$^{-1}$ (1 $\sigma$ uncertainty, period 2011–2020), accounting for 10 % of total anthropogenic $CO_2$
emissions (Friedlingstein et al., 2022). The estimated net $CO_2$ emission uncertainty ($\pm 2.6$ Gt $CO_2$ yr$^{-1}$) represents more than
50 % of the 10-year mean emission estimate and is the most uncertain emission component of the global carbon budget
(Friedlingstein et al., 2022; Houghton et al., 2012). Various sources contribute to this uncertainty, including differences in the
processes implemented in models (Bastos et al., 2020; Houghton et al., 2012; Pitman et al., 2009; McGlynn et al., 2022),
including the definition of the fluxes themselves (Pongratz et al., 2014) and the inclusion of management practices (Houghton
et al., 2012); the estimates of vegetation biomass density (Houghton, 2005); and estimates of land cover and rates of change
(Houghton et al., 2012; Bastos et al., 2021, 2020).

In support of the United Nations Framework Convention on Climate Change (UNFCCC) needs for observations of the climate
system, the Global Climate Observing System (GCOS) has identified 54 Essential Climate Variables (ECVs) that critically
contribute to improved characterization of the state of the global climate, making predictions of climate changes, and
performing attribution of the causes of such changes (GCOS, 2016). As a direct response, the European Space Agency (ESA)
launched the Climate Change Initiative (CCI) to provide stable, long-term, and consistent satellite climate data records
(Hollmann et al., 2013). The CCI thereby provides useful information to monitor the Paris Agreement goal of maintaining the
global temperature increase above pre-industrial levels to less than 2°C (UNFCCC, 2016).

Land cover, the observed biophysical cover of the Earth's surface (Di Gregorio and Jansen, 2005; Turner et al., 1993), is an
ECV (Sessa, 2008) tackled by the ESA CCI (Plummer et al., 2017). The ESA CCI medium-resolution land cover (MRLC)
dataset, operationalized within the EU Copernicus Climate Change Service (C3S) (2016-2020) thanks to strong user
endorsement, provides the longest consistent land cover climate data record, with annual maps from 1992 to 2020 at a spatial
resolution of 300 m. It describes the land surface in 22 land cover classes according to the standard of the United Nations Land
Cover Classification System (UN-LCCS) (Di Gregorio and Jansen, 2005) and 13 land cover change types consistent with the
IPCC land categories (Defourny et al., submitted).

The land surface components of global circulation models and global Earth system models play a significant role in quantifying
the historical and present-day representations of land use and land cover change impacts on climate. Most land surface models
(LSMs) parameterize global vegetation processes (e.g., photosynthesis and evapotranspiration) for a reduced set of globally
representative and similarly behaving plant types, referred to as Plant Functional Types (PFTs). PFTs can be related to
physiognomy and phenology (Anon, 1991 in Box, 1996), climate (which defines the geographical ranges in which a plant type
can grow and reproduce under natural conditions; Box, 1981), and physiological activity (e.g., $C_3/C_4$ photosynthetic pathways).

Spectral information acquired by remote sensing techniques does not allow direct mapping of PFTs. However, land cover map
series derived from satellite Earth Observations (EO) are a valuable source of physiognomy (life form and leaf type) and
phenology information for inferring the spatial distribution of PFTs. EO-derived land cover maps must be translated ("cross-
walked") into model-specific PFTs, which is typically accomplished using the information provided by the land cover class
legend (Jung et al., 2006). Differences in land cover categories, spatial resolutions, and temporal coverage between various
land cover products propagate errors to the cross-walked PFT maps and significantly contribute to uncertainties in deriving





gross primary production (GPP) and other climate-relevant variables at the regional scale (Poulter et al., 2011). To reduce
uncertainty in model ensembles, Poulter et al. (2015) proposed a standardized cross-walking framework that converts each
CCI MRLC class into pre-defined PFT fractions relevant for three leading ESMs (JULES-MOHC, ORCHIDEE-LSCE and
JSBACH-MPI) based on expert knowledge and auxiliary data. This reclassification procedure was implemented in a flexible
tool to generate other related PFT schemes required by the modelling community.

Hartley et al. (2017) used the same three ESMs to quantify the impact of uncertainties in (1) the land cover map and (2) the
cross-walking procedure on the spatio-temporal patterns of three important land surface variables: GPP, evapotranspiration,
and albedo. To disentangle the two sources of uncertainty, the modelling setup translated the plausible uncertainty ranges of
the land cover and cross-walking components into a common biomass scale. The simulations indicated that the uncertainty of
the cross-walking procedure contributed slightly more than the uncertainty of the land cover map to the inter-model uncertainty
for all three variables.

In a continuation of the ESA CCI contribution to the land cover ECV, this work aims to reduce the cross-walking component
of uncertainty by adding spatial variability to the PFT composition within a land cover class. This work moves beyond fine-
tuning the cross-walking approach for specific land cover classes and/or regions and, instead, separately quantifies the PFT
fractional composition for each 300 m pixel globally for each year in the time series (1992–2020). The new PFT product is
generated by fusing the annual CCI MRLC map series with existing high-resolution auxiliary data products that individually
characterize one surface type with high accuracy. The resulting 300 m PFT product is a companion time series of continuous
field PFT fractions that is consistent with the existing CCI MRLC map series. The global PFT product has an annual resolution,
covering 1992–2020, and indicates the specific percentage cover of 14 PFTs for each pixel at 300 m resolution. The set of 14
PFTs represented in the product includes the full set of 13 PFTs initially developed by Poulter et al. (2015) complemented
with a new built-up surface type. The full set of PFTs includes bare soil, built, water, snow and ice, natural grasses, managed
grasses (i.e., herbaceous cropland), broadleaved deciduous trees, broadleaved evergreen trees, needleleaved deciduous trees,
needleleaved evergreen trees, broadleaved deciduous shrubs, broadleaved evergreen shrubs, needleleaved deciduous shrubs,
and needleleaved evergreen shrubs. Thus, in this paper, the term "plant functional type" is applied even to the abiotic surface
types to cleanly differentiate between the land types derived from Earth observation data (i.e., land cover classes) and the land
types required by models (i.e., PFTs). Finally, these new PFT maps have been used in two land surface models (ORCHIDEE
and JULES) to demonstrate their benefit over the conventional maps based on a generic cross-walking table. For brevity, the
new PFT product is referred to as "PFT$_{local}$" due to the new localised nature of the PFT fractions at the pixel level. Products
derived by using the global cross-walking approach (using the same version 2.0.8 of the CCI MRLC map series) are referred
to as "PFT$_{global}$."

The following sections describe the auxiliary inputs and method used to quantitatively determine the PFT fractional
composition for each 300 m pixel globally; a description of the new PFT data product; and modelling results from the
application of the new PFT distribution for the year 2010 to the ORCHIDEE and JULES land surface models.
**2 Methods**
The PFT distribution was created by combining auxiliary data products with the CCI MRLC map series. The land cover
classification provides the broad characteristics of the 300 m pixel, including the expected vegetation form(s) (tree, shrub,
grass) and/or abiotic land type(s) (water, bare area, snow and ice, built-up) in the pixel. For some classes, the class legend
specifies an expected range for the fractional covers of the contributing PFTs and broadly differentiates between natural and



cultivated vegetation. The applied auxiliary data products (described in Sect. 2.1; e.g., surface water cover and tree cover) are
of higher resolution than the 300 m land cover product and therefore serve as the basis for computing the fractional covers of
the contributing PFTs at 300 m resolution. In cases of inconsistency between the land cover product and the auxiliary datasets
– for example, if the tree cover percentage derived from the auxiliary products falls outside of the range suggested by the class
legend for a 300 m pixel – the characteristics from the land cover classification are maintained. This achieves a strong coupling
between the CCI MRLC maps dataset and this new CCI PFT dataset. Deference to the class legend provides guardrails for the
temporal extrapolation of the PFT fractional covers across the entire time series (1992–2020) given the lack of available
auxiliary inputs extending across the full era. The approaches used to estimate the PFT fractions at 300 m resolution differ for
(1) pixels that did not experience a change in land cover classification over the period 1992–2020 (termed "static pixels",
described in Sect. 2.2.1) and (2) pixels that did experience a change at least once in this period (termed "change pixels",
described in Sect. 2.2.2).

### 2.1 Input datasets

#### 2.1.1 CCI medium-resolution land cover time series (300 m)

The CCI MRLC product (Defourny et al., submitted) delineates 22 primary classes and 15 additional sub-classes of land cover
at a 10-arcsecond (300 m) resolution (**Table 1**). The maps have global coverage and an annual time step extending from 1992
through 2020, with plans for the continued release of maps for 2021 and future years. The classification system used for the
CCI MRLC map series is based on the Land Cover Classification System (LCCS) of the United Nations Food and Agriculture
Organization (UN FAO) (Di Gregorio and Jansen, 2005). The LCCS defines fundamental landscape elements called
"classifiers" (e.g., trees) forming the class legend when combined in various proportions (e.g. tree cover, broadleaved,
evergreen, closed to open (>15 %)). The 15 sub-classes, also called "regional classes," are defined only in geographic regions
where appropriate training data is available and are those with a numeric classification code that has a final digit of 1, 2, or 3
(**Table 1**). The 22 primary classes and 15 sub-classes are collectively referred to here as simply "classes." For each year of the
time series, each 300 m pixel in the dataset is assigned as a single land cover class. The change detection algorithm monitors
thirteen possible land cover transitions through time. For a pixel to register a change in its assigned land cover class, the
algorithm must identify the change for two consecutive years in the workflow. A lack of change in a pixel's assigned class
does not necessarily indicate an absence of change in the land surface over the time series; rather, it indicates that any change
that has occurred in the pixel was limited enough in scale or duration that the assigned class did not change. The full time
series and an associated set of quality flags are freely available at https://maps.elie.ucl.ac.be/CCI/viewer/ (last access August
2022) in GeoTiff and https://cds.climate.copernicus.eu/cdsapp#!/dataset/satellite-land-cover?tab=overview (last access
August 2022) in netCDF. This CCI PFT product is based on v2.0.8 of the CCI MRLC time series, which includes corrections
for the known overestimation of cropland relative to grassland in South America (Defourny et al., submitted).

#### 2.1.2 Surface water product (30 m)

The Landsat-based surface water product developed by the Joint Research Centre (Pekel et al., 2016) is used to derive the
permanent inland water fractions at 300 m resolution (calculation details in Sect. 2.2). The surface water occurrence layer
(obtained at https://global-surface-water.appspot.com) indicates the frequency of water occurrence in each 30 m pixel (80°N–
60°S) over the period March 1984 to December 2019. The frequency occurrence data is reported as integer values of 1–100
%, where a value of 100 % occurrence indicates a permanent water surface that existed over the entire analysis period, which
encompasses all but the most recent year (2020) of the time series of the MRLC product.



### 2.1.3 Tree canopy cover product (30 m)

A Landsat-based tree canopy cover product (Hansen et al., 2013) is used to derive the tree cover fractions for 300 m pixels belonging to vegetated classes (except where otherwise noted in Sect. 2.2). The product (obtained at https://glad.umd.edu/Potapov/TCC_2010/) is based on the application of a regression tree model to growing-season Landsat 7 ETM+ data (https://glad.umd.edu/dataset/global-2010-tree-cover-30-m). The dataset indicates the maximum tree canopy cover percentage (integer values of 1–100 %) at 30 m resolution (80°N–60°S) and is approximately representative of 2010.

### 2.1.4 Tree canopy height product (30 m)

The global forest canopy height product from Potapov et al. (2021) is used to derive the fractional covers of trees and shrubs in 300 m pixels classified as shrubland. The 30 m product (obtained at https://glad.umd.edu/dataset/gedi/) was created by combining the footprint-level lidar forest height measurements (using the 95$^{th}$ percentile relative height metric) for April–October 2019 from the Global Ecosystem Dynamics Investigation with wall-to-wall Landsat optical data to perform spatiotemporal extrapolation. The resulting dataset indicates the canopy height (0–60 m) at 30 m resolution (52°N–52°S), where canopy heights < 3 m were set to 0 m under the assumption that the pixel lacks woody vegetation.

### 2.1.5 Built-up product (38 m)

The Landsat-based Global Human Settlement Layer (GHSL) dataset produced by the Joint Research Centre (Pesaresi et al., 2013) is used to derive the built-up fraction for 300 m pixels classified as urban land cover by the Global Urban Footprint (GUF) dataset (Esch et al., 2017). The built-up fraction of the PFT dataset is defined as buildings, roads, and man-made structures. The GHSL (alpha version dated November 2014) consists of globally consistent built-up maps for four consecutive years (1975, 1992, 2000, and 2014) at 38 m resolution. Built-up areas include both permanent and temporary above-ground buildings.

### 2.1.6 Zonation products

In addition, three zonation products are used complementarily to consolidate the assignment of the phenology type (deciduous or evergreen) and leaf type (broadleaved or needleleaved) to shrubs and, in a very small number of pixels, to trees belonging to a class legend of mixed trees. The Köppen-Geiger climate zone product from Beck et al. (2018) divides the Earth's land surface into 30 distinct climate zones at 0.0083° resolution (about 1 km) based on present-day (1980–2016) temperature and precipitation records. Data were obtained at https://figshare.com/articles/dataset/Present_and_future_K_ppen-Geiger_climate_classification_maps_at_1-km_resolution/6396959/2. The landform dataset from Sayre et al. (2014) identifies landforms – surface water, plains, hills, or mountains – at 250 m resolution for 83.6°N–56°S and is derived from a digital elevation model (USGS GMTED2010: Danielson and Gesch, 2011). The data product was obtained at https://rmgsc.cr.usgs.gov/outgoing/ecosystems/Global/. Finally, world regions follow the definitions used in the Integrated Model to Assess the Global Environment 3.0 (IMAGE03) (Stehfest et al., 2014). The IMAGE03 regional classification framework has been harmonized with the CCI MRLC grid by reconstructing the original dataset using the IMAGE-based list of countries per region (available at https://models.pbl.nl/image/index.php/Region_classification_map) along with country boundaries from the FAO Global Administrative Unit Layers (available at https://data.apps.fao.org/), expanding the list to include Antarctica, Greenland, and additional small islands. The resulting raster dataset divides Earth's surface into 28 regions on the CCI MRLC grid.

### 2.1.9 CCI medium-resolution water body product

The CCI MRLC water body product (Lamarche et al., 2017) is used to delineate between inland water and ocean. The dataset (available at http://maps.elie.ucl.ac.be/CCI/viewer/download.php) designates all pixels at 150 m resolution as either ocean or



non-ocean, the latter of which includes both land and inland water. The dataset is consistent with the water body class (code
210) of the land cover maps of the CCI MRLC. An updated version 4.1 of the product was used here, in which the North
American Great Lakes are now considered to be inland water rather than ocean. It is available at
http://maps.elie.ucl.ac.be/CCI/viewer/download.php.
**2.2 PFT dataset development**
The overall approach assumes that the definition of the MRLC class is the basis for harmonizing the four existing high-
resolution land cover data sets. It proceeds through a systematic sequence of estimating water fraction and tree cover fraction,
using tree height to assign life form, and finally deriving phenology. This step-by-step approach is first applied to static pixels
before extending it to pixels that undergo changes over time, as identified in the CCI MRLC map series.
**2.2.1 Static pixels**
For static pixels – that is, pixels that have not experienced a class change over the era covered by the CCI MRLC time series
(1992–2020) – the derived PFT fractions are treated as temporally invariant for the entire period. Therefore, any intra-pixel
change in the fractions of a static pixel is not captured in the PFT map series due to a lack of temporally resolved auxiliary
inputs extending over the full time series. Such a change is expected to be so limited in scale and/or duration that it did not
prompt a change in class assignment, underscoring the appropriateness of treating the fractional composition of the static pixels
as consistent over time.

The same set of auxiliary inputs and the same calculation method are applied to the widest possible set of land cover classes
to ensure spatial consistency in the derived PFT fractions. Nonetheless, inherent differences between the classes necessitate
the use of different input datasets and methods in some cases. For each class, only a subset of the 14 PFTs is permitted non-
zero fractions (**Table 1**). Because the PFT fractional composition is estimated independently for each 300 m pixel of a class,
in some cases, an individual pixel of the class can have zero fractional cover even for a PFT that is allowed non-zero cover for
that class. For all pixels, the sum of PFT fractions is 100 %.

The 30 m water frequency occurrence dataset of Pekel et al. (2016) is used to estimate the permanent inland water fraction of
the 300 m pixels for all but the permanent snow and ice class, which has no liquid surface water cover. A threshold of 90 %
frequency occurrence is applied to assign 30 m pixels as either water (frequency occurrence $\geq$ 90 %) or non-water (frequency
occurrence < 90 %). The resulting binary representation of water/non-water is aggregated to 300 m to estimate the percentage
of the 300 m pixel that is permanent inland water PFT.

The percentage of the 300 m pixel that is vegetated is calculated as 100 % minus the inland water percentage; that is, for all
vegetation-containing classes except for the sparse vegetation classes, which have bare soil PFT cover, all non-inland-water
area in the 300 m pixel is entirely vegetated (0 % bare soil PFT) in the PFT product. Pixels belonging to the shrubland classes
(codes 120–122 and 180) can have a mixture of trees, shrubs, and herbaceous cover. For pixels of non-shrubland vegetation-
containing classes, the vegetated portion of the pixel is composed of trees and herbaceous cover (i.e., cropland and/or natural
grass). The percentage of the 300 m pixel that is tree cover is estimated using the 30 m tree cover dataset for 2010 from Hansen
et al. (2013). This Landsat-based dataset provides the percentage of tree canopy cover (integers 1–100 %) based on growing
season observations. The tree cover percentage of the vegetated (i.e., non-water) portion of the 300 m pixel is obtained from
the median of the tree canopy cover fractions of the non-water 30 m pixels, where the 30 m non-water pixels are identified
using the binary water/non-water representation derived using the surface water occurrence dataset. The tree cover percentage
of the entire 300 m pixel is calculated as the product of this value (the tree cover fraction of the non-water part of the grid cell)



and the non-water fraction of the grid cell. This approach harmonizes the Landsat-based surface water occurrence and tree
canopy cover datasets such that the combined tree and water percentages never exceed 100 %.

For the tree cover classes 50–82, the class legend specifies an expected range for the tree cover percentage (**Table 1**, class
description column). For the tree cover classes 90, 160, and 170, a tree cover fraction of >15 % is implicit from the UN LCCS.
Based on the spatial and temporal consistency of the map series, deference is made to the class legend for pixels in which the
estimated tree cover fraction derived from the auxiliary datasets disagrees with the class legend. This allows the PFT product
to retain the advantages of the CCI MRLC map series while improving the translation of the land cover dataset into PFT maps.
For tree cover class pixels in which the estimated tree cover fraction derived from the auxiliary datasets disagrees with the
class legend, the mean tree cover among all static 300 m pixels of its class is calculated over the 0.25° longitude × 0.25°
latitude window overlapping the pixel – that is, a window with width and height of 0.25° with the pixel of interest at the centre.
The mean is based on the initially calculated tree cover fractions derived from the auxiliary data products (i.e., the tree cover
fraction harmonized with the surface water occurrence dataset). The window is expanded to 0.5° longitude × 0.5° latitude if
no static pixels of the class exist in the smaller window. (Because class 82 has so few pixels globally, class 72 pixels are
additionally applied in the window mean calculation for class 82 pixels.)

One of five cases is possible:
(1) If the mean tree fraction for the window falls within the expected range based on the class legend, then the tree cover
fraction of the pixel of interest is assigned as the mean tree fraction for the window.
(2) If the mean tree fraction for the window is higher than the upper limit specified by the class legend, then the tree cover
fraction of the pixel of interest is assigned as the upper limit from the legend.
(3) If the mean tree fraction for the window is lower than the lower limit specified by the class legend, then the tree cover
fraction of the pixel of interest is assigned as the lower limit from the legend.
(4) If a window of 0.5° × 0.5° does not have any pixels of the class of interest and the tree cover fraction derived from the
auxiliary products exceeds the upper limit specified by the class legend, then the tree cover fraction for the pixel is
assigned as the upper limit of the class legend.
(5) If a window of 0.5° × 0.5° does not have any pixels of the class of interest and the tree cover fraction derived from the
auxiliary products is lower than the lower limit specified by the class legend, then the tree cover fraction for the pixel is
assigned as the lower limit of the class legend.

For pixels that belong to a tree cover class and had tree cover percentages assigned using the neighbourhood mean, the resulting
sum of the inland water and tree cover percentages can exceed 100 %. In such cases, the tree cover percentage is calculated as
100 % minus the inland water percentage. If the resulting tree cover percentage is lower than the legend minimum for that
class, then the tree cover percentage is set as the legend minimum and the water percentage is set as the residual area in the
pixel (100 % minus tree cover percentage). For all tree cover class pixels, the grass cover percentage is calculated as 100 %
minus the final tree cover percentage, and the grass type is assigned as natural grasses. No minimum water percentage is
defined for the flooded tree cover classes (codes 160 and 170).

For the biotic classes rainfed cropland (codes 10, 11, 12), irrigated or post-flooding cropland (code 20), mosaic of cropland–
natural vegetation (codes 30 and 40), mosaic of woody–herbaceous vegetation (codes 100 and 110), and grassland (code 130),
the tree cover percentage derived from the auxiliary products is used directly since the legend does not specifically define the
expected tree cover; therefore, modification of the PFT fractions based on the class legend is not applied for these classes as it
is for some other classes. The percentage of the 300 m pixel that is grass cover is calculated as 100 % minus the sum of the
inland water and tree cover percentages. The grass type – managed (i.e., crops) or natural – is defined by the class legend. For
most mixed classes, the assigned grass type reflects the majority type as indicated by the legend. All grass in the pixel is
assigned as managed grass for classes 10, 11, 12, 20, and 30. Pixels belonging to the mosaic class 40 have a mix of herbaceous



crops (up to 49 % of the pixel area) and natural grasses (for excess grass cover beyond 49 % of the pixel area). All grass cover
is assigned as natural grass for all other classes.

In some of the classes in this set, an expected percentage cover is given for total woody vegetation (trees and shrubs) or for
the shares of cropland and natural vegetation, where the two categories differentiate by management status rather than life
form. In the PFT product, shrub cover is estimated only for the shrubland classes due to a lack of appropriate auxiliary inputs
to discriminate between trees and shrubs for all classes, so modification of the life form shares in such pixels based on the
legend description may introduce additional bias in the PFT product and is therefore avoided. Management status (cropland
vs natural) is assigned in the PFT product only for grasses and is based on the class descriptions, so an independent assessment
of the shares by management status is not possible.

Pixels belonging to the sparse vegetation classes (codes 150, 151, 152, and 153) can have non-zero fractions of bare soil, trees,
natural grass, and inland water. The class definition requires a vegetation fraction of 4–14 %. Since shrub cover is not estimated
for the sparse vegetation classes, the vegetation component is composed of trees and natural grasses; therefore, the total
vegetation fraction is enforced for sparse vegetation pixels, but the resulting life form may differ from that indicated by the
legend for the sub-classes with codes 151–153. If the tree cover derived from the auxiliary inputs is ≥ 15 %, then the tree PFT
is reduced to 14 % in deference to the legend of the CCI MRLC map series, natural grass PFT is assigned as 0 % since tree
cover accounts for the maximum total vegetation fraction (trees + grass), and the bare soil PFT percentage is calculated as 100
% minus the inland water percentage minus 14 % tree PFT. If the tree cover derived from the auxiliary inputs is < 15 %, then
this input tree percentage value is assigned as the final tree PFT percentage in the pixel and additional legend-consistency steps
are applied to assign the grass and bare fractions: (1) if the non-water area of the pixel is 4–14 %, then natural grass PFT
accounts for the residual portion of the pixel (14 % minus tree PFT percentage minus inland water percentage); (2) if the non-
water percentage of the pixel is < 4 %, then the natural grass PFT percentage is calculated as 4 % minus the tree PFT percentage
(since the lower bound on total vegetation is 4 %) and the water PFT percentage is scaled down to 96 %; or (3) if the non-
water percentage of the grid cell exceeds 14 %, then the natural grass percentage is calculated as 14 % minus the tree PFT
percentage (that is, the upper bound of 14 % is assumed for total vegetation cover) and the residual pixel area is assigned as
bare soil PFT (100 % minus water PFT percentage minus 14 % vegetation cover).

A mixture of tree and shrub woody vegetation types is assigned to pixels of the shrubland classes (codes 120, 121, 122, and
180). The 30 m resolution tree canopy height dataset from Potapov et al. (2021) is applied to discriminate between shrubs and
trees in pixels that are covered by this data product (52°N–52°S). Potapov et al. (2021) re-assign pixel values of ≤ 2 m to 0 m
height. Here, the 30 m resolution pixels are assigned to three broad height classes: 0 m, 3–5 m, and > 5 m. Mean re-sampling
to the 300 m resolution of the land cover dataset results in pixel values that indicate the percentage cover of the three height
classes. The percentage cover of the 3–5 m height class is taken to be the percentage shrub cover in the 300 m pixel and the
percentage cover of the > 5 m height class is taken to be the percentage tree cover in the 300 m pixel, recognizing that there
may be some bias introduced by 30 m pixels in the input dataset that contain both shrubs and trees. In deference to the class
legend, 300 m pixels with shrub cover < 16 % are assigned as having 16 % shrub cover and those with tree cover > 15 % are
assigned as having 16 % tree cover. For shrubland pixels that occur outside of the extent of the Potapov et al. (2021) data
product (52°N–52°S), the tree cover percentage is assigned according to the tree cover input derived from Hansen et al. (2013)
and the shrub cover percentage is assigned following the most recent version of the global cross-walking table (CWT) (60 %
shrub cover for classes 120–122 and 40 % shrub cover for class 180). For all shrubland pixels, in cases where the sum of water,
tree, and shrub cover exceeds 100 %, the three PFTs are scaled down proportionally so that the sum is 100 % while retaining
the legend expectations for the tree and shrub cover. Natural grass cover is assigned as the residual area of the pixel in cases
where the sum of water, tree, and shrub cover is < 100 %. No minimum water percentage is defined for the flooded shrubland
class (code 180).

Pixels classified as urban (code 190) can have non-zero fractions of inland water, trees, natural grass, and urban impervious
(built-up) PFTs. In the land cover classification, pixels are assigned as an urban class when a minimum threshold of 50 % built
was exceeded based on the GUF dataset (Esch et al., 2017). In the PFT product, the tree and surface water fractions are derived
using the same protocol as the one applied to the vegetated classes. The urban impervious fraction is derived from the GHSL
dataset (Pesaresi et al., 2013) by aggregating the built-up pixels from the four epochs into a binary built-up / non-built-up
distribution at 38 m. Re-sampling to 300 m provides the percentage of the 300 m pixel that is built PFT, introducing local
variability which at the global scale ranges from 0–100 % built. Only pixels classed as urban by GUF are assigned a non-zero
urban impervious fraction in the PFT dataset. Non-urban pixels (i.e., those with less than 50 % urban land cover according to
GUF) are not refined with GHSL data or assigned a percentage built-up. The GHSL appears to capture urban impervious areas
more consistently whereas GUF misses road fractions in the built fractions. This is most notable in rural areas and a few
selected locations in city centres. If the sum of the urban impervious, tree, and water fractions exceeds 100 %, then the urban
impervious percentage is retained and the water and tree percentages are scaled down proportionally to a total sum of 100 %;
otherwise, the residual of the urban impervious, tree, and water percentages is assigned as the natural grass percentage.

Water body class (code 210) pixels that are ocean are assigned as 100 % water PFT, while those that are inland can additionally
have a non-zero cover of tree and natural grass PFTs. The designation of ocean vs. inland at 300 m is determined using the
150 m water body product. The ocean designation is applied to water body class pixels in which all four of the overlapping
150 m pixels of the water body product are classified as the ocean; all other water body class pixels are designated as inland
water. The water and tree PFT fractions for inland water body class pixels are assigned using the same 300 m harmonized
surface water and tree cover auxiliary inputs that are used for the other classes; however, a minimum of 86 % water PFT is
enforced following the legend definition for this class. If the sum of the tree fraction and the adjusted water PFT fraction
exceeds 100 %, then the tree percentage is scaled down as 100 % minus the adjusted water PFT percentage. Any residual area
is assigned as natural grass PFT.

The bare area classes (codes 200, 201, and 202) can have up to 3 % vegetation cover, by definition, so bare area pixels can
have non-zero fractions of bare soil, tree, and water PFTs. The auxiliary products define the tree and inland water fractions,
but tree cover exceeding 3 % is scaled down to the class maximum of 3 %. Bare soil PFT percentage is calculated as 100 %
minus the inland water percentage minus the tree percentage. Pixels of the mosses and lichens class (code 140) can have non-
zero fractions of surface water and natural grasses, the latter of which is estimated as 100 % minus the inland water percentage.

The permanent snow and ice class (code 220) is assigned as 100 % snow and ice PFT. All other classes are assigned as 0 %
snow and ice PFT. Nearly all pixels classified as permanent snow and ice class in the CCI MRLC time series are static pixels;
that is, such pixels are snow and ice cover for every year of the land cover map series. This is due to a lack of temporally
resolved input data available at the global scale to track the evolution of this surface type. Therefore, neither the CCI MRLC
classification nor the associated PFT product should be used to track changes in glaciers over time.

For all pixels – of any class – that have a non-zero tree fraction, the total tree fraction is assigned as a single tree type
(broadleaved or needleleaved leaf type, deciduous or evergreen phenology). For the tree cover classes coded 50–82, the specific
tree type follows the class legend. For example, class 50 is defined as "Tree cover – broadleaved evergreen >15 %," so the tree
component of this class is assigned as the broadleaved evergreen tree type. Tree cover is assigned as broadleaved deciduous



in pixels of classes 60–62, needleleaved evergreen in pixels of classes 70–72, and needleleaved deciduous in pixels of classes
80–82. For pixels of the tree cover classes coded 90, 160, and 170 and all other vegetation-containing classes except the
shrubland classes, the specific tree type is assigned by pixel based on the majority tree type in the surrounding $0.25° \times 0.25°$
neighbourhood window, where the majority calculation is performed on static pixels of the tree cover classes with legend-
defined tree types (classes 50–82). If the $0.25° \times 0.25°$ window does not contain any static pixels of the well-defined tree types,
then the window is incrementally expanded by $0.25°$ in each direction (longitude and latitude) to a maximum window size of
$2° \times 2°$ until such a pixel is contained within the search window. The same tree type is assigned to all pixels in a class for the
tree cover classes 50–82, while the assigned tree type can vary between pixels within a class for the other classes.

For a very small number of pixels, static pixels of the type-defined tree cover classes are absent from the surrounding $2° \times 2°$
window, so a climatological approach is instead used to assign the tree type to such pixels. This approach uses three auxiliary
inputs: (1) the present-day Köppen-Geiger climate zone map from Beck et al. (2018), downscaled from 1 km resolution to the
300 m CCI MRLC grid using mode resampling; (2) the map of world regions derived for use with the IMAGE03 model,
expanded to include Greenland, Antarctica, and additional small islands; and (3) the landform map from Sayre et al. (2014),
resampled from 250 m resolution to the 300 m CCI MRLC grid using mode resampling. A nearest neighbour analysis is used
to gap fill missing data at 300 m resolution for each of the three auxiliary inputs. Pixels requiring data are those with < 100 %
water PFT cover in the PFT product. Pixels that are designated as surface water in the landform dataset and have < 100 %
water PFT cover in the PFT product are additionally filled with one of the terrestrial landforms (plains, hills, and mountains).
Missing data generally occur along coastlines due to mismatches in the land-sea masks of the auxiliary datasets and the CCI
MRLC data. The gap-filled datasets are combined to create a dataset of 1,531 unique combinations of landform, region, and
climate zone. For each of the unique combinations, the areal cover of each of the tree cover classes with well-defined tree
types (codes 50–82) is calculated using static pixels of those classes, and the majority tree type by area is identified for each
unique combination. There are very few static pixels of the type-defined tree cover classes in the Middle East and Sahara
regions, so the dominant tree type in these regions is set as broadleaved deciduous. For pixels in which the tree type –
broadleaved or needleleaved, deciduous or evergreen – cannot be assigned based on the neighbourhood window, the majority
tree type of the pixel's unique zone is assigned. This method is also applied to assign the types of both shrubs and trees in all
shrubland class pixels. Thus, there may be inconsistencies between the shrub type indicated by the class legend and that
assigned using this biogeographical approach.

Most of the auxiliary inputs are based on Landsat images and therefore have an extent of 80°N–60°S. The main processing
algorithm for the PFT product, explained above for the static pixels, therefore operates on this extent. Less than 0.5 % of the
area outside of this extent is composed of pixels belonging to a class other than water bodies (code 210) or permanent snow
and ice (code 220). The largest contributors to this small area are the sparse vegetation classes followed by the bare area classes
with negligible contributions from shrubland (including flooded shrubland), grassland, and lichens and mosses classes. To
extend the PFT product to global extent, the following assumptions are applied to the pixels north of 80°N and south of 60°S:
(1) 100 % snow and ice PFT is assigned to pixels of the permanent snow and ice class; (2) 100 % water PFT is assigned to
pixels of the water body class; (3) 100 % bare soil PFT is assigned to pixels of the bare area classes; (4) 100 % natural grass
PFT is assigned to pixels of the grassland and lichens and mosses classes; (5) 96 % bare soil PFT and 4 % natural grass PFT
(to meet the legend minimum of vegetation cover) are assigned to pixels of the sparse vegetation classes; and (6) 84 % natural
grass PFT and 16 % needleleaved deciduous shrub PFT (matching the legend minimum shrub cover) are assigned to pixels of
the shrubland classes. For the shrubland classes, the shrub type of needleleaved deciduous is assigned because the shrubland
class pixels needing assignment (north central Russia) occur nearest pixels of needleleaved deciduous shrubs that had shrub
type assigned using the standard method.



### 2.2.2 Pixels experiencing land cover change

Dynamic pixels – that is, pixels that have experienced at least one land cover class change over the 1992–2020 era – correspond to 5.88 % of the ice-free land surface (Defourny et al., submitted). For such pixels, the derived PFT fractions are derived for each of the classes assigned to that pixel over the era. For example, if a pixel changed from forest to cropland, PFT fractions associated with the forest class are estimated and PFT fractions associated with the cropland class are also estimated for the pixel. The method used to assign the PFT fractions depends on the time stamp of the class in relation to the time stamp (2010) of the auxiliary dataset (Hansen et al., 2013) from which the tree cover fractions are derived. The PFT fraction of a pixel in 2010 was derived using the following class-specific methods described in Sect. 2.2.1. Any change of class occurring before or after 2010 leads to deriving new PFT fractions as the mean PFT fractions of all 300 m pixels of the same class pixels within the overlapping 0.25° x 0.25° window centred on the pixel of interest. The input pixels over which the mean is calculated are the 300 m pixels that did not experience land cover class change over the 1992–2020 era. If no pixels of the relevant class are within the 0.25° x 0.25° window, then the window is incrementally expanded by 0.25° in both the latitude and longitude directions until at least one pixel of the relevant class is contained in the window. A pixel can experience up to 7 land cover changes in the 1992–2020 era (Defourny et al., submitted), which leads to deriving new PFT fractions for each new land cover class encountered.

### 2.3 Modelling assessment

The impact of the updated PFT distribution on land surface fluxes is evaluated using global simulations of two land surface models: the Organizing Carbon and Hydrology in Dynamic Ecosystems (ORCHIDEE; Krinner et al. 2005 and later revisions) and the Joint UK Land Environment Simulator (JULES; Best et al., 2011; Clark et al., 2011). The simulations with ORCHIDEE focus on evaluating the impact of the updated PFT distributions on a selected set of climate-relevant variables. The ORCHIDEE model applies the Climatic Research Unit (CRU)–Japanese reanalysis (JRA55) v2.0 6-hourly atmospheric driving data for 1901–2018 (Harris et al., 2014; Kobayashi et al., 2015; UEA CRU and Harris, 2019) and the CCI PFT distribution maps for 2010. Two PFT distributions are applied: (1) the new PFT map (PFT$_{local}$) described above and (2) the PFT distribution based on the application of the global standard CWT to the CCI MRLC product for 2010 (PFT$_{global}$) (Table C1) (Hartley et al., 2017, Lurton et al., 2020). The 2010 PFT map is used (recycled) for each year of the simulation. ORCHIDEE is run at a horizontal resolution of 0.5° latitude $\times$ 0.5° longitude over the period 1900 - 2018, and all simulated data before 1980 are discarded as spin-up, with analysis based on the years 1980–2018. The impact of the updated distribution relative to that based on the global CWT is compared with ORCHIDEE for an ensemble of climate-related variables, including albedo, surface fluxes (latent and sensible heat and their ratio), gross primary productivity, surface temperature, tree fraction, leaf area index (LAI), and above-ground biomass (Sect. 4).

In a separate assessment of the implications of the updated PFT distributions for model evaluation, JULES simulations of PFT distributions, created for the Inter-Sectoral Impacts Model Inter-comparison Project (ISIMIP; Frieler et al., 2017), were used. This was done to compare evaluation results using both the CWT-derived PFT distributions (PFT$_{global}$) and the updated PFT distributions (PFT$_{local}$). The 2010 PFT distributions are used to evaluate the JULES dynamic vegetation results. JULES was driven by the ISIMIP2b protocol described in Frieler et al. (2017) and applied to JULES as described in Mathison et al. (in preparation). Sect. 4.2 describes the Dynamic Global Vegetation Model results for 2010 from the JULES offline simulations driven by HADGEM2-ES climate for the period 1850 to 2100.



# 3 CCI PFT dataset description

## 3.1 General description

The CCI PFT dataset (hereafter called PFT$_{local}$) provides the percentage cover as discrete values of 0–100 % of 14 PFTs at 10 arc-second resolution (300 m at the equator; 64,800 pixels in the latitude dimension × 129,600 pixels in the longitude dimension). The global continuous field maps are produced at an annual resolution, covering the years 1992–2020. The PFT distributions are consistent with the CCI MRLC data product and eliminate the need to use a CWT to translate land cover classes into PFTs. The 14 PFTs encompass: (1) permanent inland water bodies; (2) permanent snow and ice cover; (3) bare soil; (4) built-up areas, which includes artificial impervious area such as buildings and, frequently but not exhaustively, other paved surfaces such as roads; (5) managed grasses (i.e., herbaceous crops); (6) natural grasses (i.e., non-cultivated herbaceous vegetation); (7) broadleaved deciduous shrubs; (8) broadleaved evergreen shrubs; (9) needleleaved deciduous shrubs; (10) needleleaved evergreen shrubs; (11) broadleaved deciduous trees; (12) broadleaved evergreen trees; (13) needleleaved deciduous trees; and (14) needleleaved evergreen trees (Figure 1). Following the auxiliary inputs, trees are woody vegetation with a height > 5 m, while shrubs are woody vegetation with a height of 3–5 m, inclusive. An updated water body product (version 4.1) at 150 m resolution, used here to delineate between inland water and ocean, likewise replaces the older version and can be downloaded from the same data repository as the PFT maps.

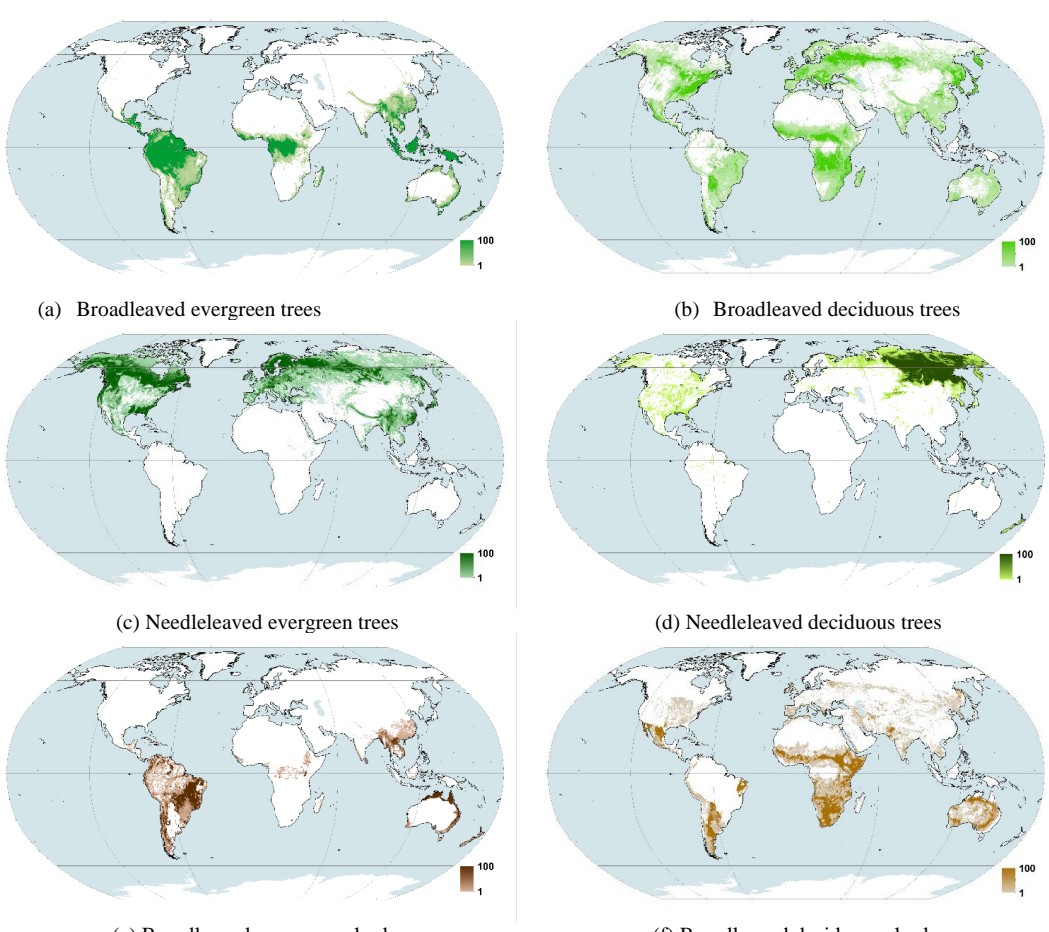

(a) Broadleaved evergreen trees

(b) Broadleaved deciduous trees

(c) Needleleaved evergreen trees

(d) Needleleaved deciduous trees

(e) Broadleaved evergreen shrubs

(f) Broadleaved deciduous shrubs

(g) Needleleaved evergreen shrubs

(h) Needleleaved deciduous shrubs

(i) Managed grasses

(j) Natural grasses

(k) Built-up areas

(l) Permanent inland water bodies

(m) Bare soil

(n) Permanent snow and ice cover

**Figure 1. Percentage cover in 2010 for the 14 PFTs included in the** PFT$_{local}$ data product at a spatial resolution of 0.25° × 0.25°. (a) Broadleaved evergreen trees, (b) Broadleaved deciduous trees, (c) Needleleaved evergreen trees, (d) Needleleaved deciduous trees, (e) Broadleaved evergreen shrubs, (f) Broadleaved deciduous shrubs, (g) Needleleaved evergreen shrubs, (h) Needleleaved deciduous shrubs, (i) Managed grasses, (j) Natural grasses, (k) Built-up areas, (l) Permanent inland water bodies, (m) Bare soil, and (n) Permanent snow and ice cover.

The PFT$_{local}$ dataset indicates that herbaceous vegetation covers 44.8 % of the Earth's land surface, with around one-third of that area devoted to herbaceous crops. Tree cover accounts for 21.3 % of the land surface, which is much larger than that of shrubs (3.2 %). The abiotic surface types cumulatively cover 30.8 % of the land surface: 18.4 % bare soil, 10.0 % snow and ice, 2.1 % inland water, and 0.3 % built.



The CCI PFT dataset is provided as a companion product to the ESA CCI LC map series products with similar specifications
with a global extent, a pixel size of 300 m and a Plate Carrée projection. However, climate models may need products
associated with a coarser spatial resolution, over specific areas (e.g., for regional climate models), and/or in another projection.
To tackle the variety of requirements, a user tool has been developed that allows users to adjust the products in a way which
is suitable to their models. A minimum list of possibilities in terms of spatial resolution and projection has been established
and the conversion of CCI-Land Cover classes to other user-defined classes is also foreseen. The CCI PFT product and the
user tool are freely available at maps.elie.ucl.ac.be/CCI/viewer/ and climate.esa.int/en/projects/land-cover/data/.

### 3.2 PFT layer description considering the CCI MRLC categories and the PFT$_{global}$ dataset

Table 2 shows the global areal coverage of each PFT by class for 2010 for the PFT$_{local}$ product, and Table A1 shows the
equivalent data corresponding to the application of the most recent version of the CCI MRLC global CWT (Lurton et al. 2020;
Table A2) to the v2.0.8 CCI MRLC map for 2010 (hereafter called PFT$_{global}$). Figure A2 complements Table A1 by illustrating
the differences between the PFT$_{local}$ and the PFT$_{global}$ products globally at a spatial resolution of 0.25 x 0.25 degrees. For each
class of PFT$_{global}$, the global CWT specifies the fractional composition of contributing PFTs; in this approach, each pixel of a
class is assigned the same fractional PFT composition regardless of its location on Earth. Table 3 indicates the percentage PFT
composition by class for 2010 for PFT$_{local}$, calculated as an area-weighted mean taken over all pixels of the class globally.
Figure A3 provides a spatialized summary of the largest differences between the PFT$_{local}$ and the PFT$_{global}$ products. (a) PFTs
with the largest increase and (b) corresponding fraction gained, (c) PFT loss and (d) corresponding fractions lost are illustrated
globally with 0.25° x 0.25° pixels.

### Tree cover

The PFT$_{local}$ product indicates global areal tree cover of 31.4 million km$^2$ (Figure 1): 45.2 % broadleaved evergreen; 24.3 %
needleleaved evergreen; 23.0 % broadleaved deciduous; and 7.5 % needleleaved deciduous. The PFT$_{local}$ product indicates
global areal tree cover that is 4.6 % higher than in the PFT$_{global}$ distribution. Globally, tree coverage is higher in the PFT$_{local}$
product relative to the PFT$_{global}$ distribution for all tree types except needleleaved deciduous trees. Compared to the global
CWT, in which every pixel belonging to a given class is assigned the same PFT fractions, the updated method for estimating
PFT fractions locally results in greater variability of tree fractions among 300 m pixels within a single class. For example, the
global CWT suggests that all class 10 (rainfed cropland) pixels are 0 % tree cover, but the PFT$_{local}$ product based on auxiliary
inputs suggests a much wider range of tree cover at the pixel level, ranging from 0–100 % tree cover at the 300 m pixel level.
The distribution for Africa is shown in Figure A1, where tree crops in the Sahel are readily apparent. Globally, class 10 pixels
have 5.1 % tree cover on average (Table 3). On average, class 12 pixels (rainfed cropland – tree or shrub cover) have 18.4 %
tree cover. The auxiliary dataset used to derive tree cover for most classes in the PFT$_{local}$ product is based on Landsat 7 images
(Hansen et al. 2013); the artifacts associated with the failure of the Landsat 7 Scan Line Corrector (Andrefouet et al., 2003)
are visible in the 300 m PFT$_{local}$ dataset in some regions, particularly in west-central Africa.

### Shrub cover

The PFT$_{local}$ product indicates 4.7 million km$^2$ of global shrub cover. The largest contributors to total shrub cover are
broadleaved deciduous (44.6 %) and needleleaved evergreen shrubs (25.2 %). Shrub cover is 74 % lower in the PFT$_{local}$ product
than in the PFT$_{global}$ dataset. Some of this difference arises because the PFT$_{local}$ product estimates lower shrub PFT in shrubland
class pixels (codes 120–122 and 180) compared to the PFT$_{global}$ dataset, which estimates 8.8 million km$^2$ of shrub PFT in
shrubland classes. The area-weighted mean percentage composition of shrubs in shrubland class pixels is 30.0 % for class 120
in the PFT$_{local}$ product, 26.1 % for class 121, 34.4 % for class 122, and 30.7 % for class 180. The CWT suggests 60 % shrub
cover for classes 120–122 and 40 % for class 180. The CWT estimates 0 km$^2$ of tree PFT cumulatively in these classes





compared to 630,000 km$^2$ in the PFT product. The uncertainty associated with the height estimation in the global canopy height
product of Potapov et al. (2021) may contribute to the confusion of shrubs and trees in some cases. Nonetheless, the evidence-
based PFT$_{local}$ product indicates a significantly lower estimate for global woody vegetation cover in pixels of the shrubland
classes compared to the PFT$_{global}$ dataset, which was largely based on expert knowledge.

In addition to the differences in the shrubland class pixels, a large part of the difference in total shrub cover between the PFT$_{local}$
product and the PFT$_{global}$ dataset can be ascribed to the fact that the PFT$_{local}$ product estimates shrub PFT only in pixels
belonging to the shrubland classes (codes 120–122 and 180) due to a lack of appropriate datasets to apply to the other classes.
The CWT estimates 9.5 million km$^2$ of shrub cover in non-shrubland PFTs, and some of this shrub cover may indeed be
missing from the PFT$_{local}$ product. However, because the PFT$_{local}$ product, which is based on quantitative estimation using
auxiliary inputs, and the CWT, which is largely based on expert input, differed so strongly in the estimates of shrub PFT in
the shrubland class pixels, some of the differences in the non-shrubland class pixels may likewise be due to bias in the CWT.
**Natural and managed grasses**
Global grass PFT cover in the PFT$_{local}$ product is 65.7 million km$^2$, two-thirds of which is natural grass. Total grass cover is
29.6 % higher in the PFT$_{local}$ product than in the PFT$_{global}$ map (38.3 % higher for natural grass and 14.7 % higher for managed
grass). In the PFT$_{local}$ product algorithm, for the vegetated classes except for sparse vegetation, the entire non-water fraction
of the 300 m pixel is assigned as vegetation; typically, water, trees, and other PFTs are estimated based on auxiliary inputs and
the CCI MRLC class legend, and then the residual area is assigned as grass cover. Thus, grass vegetation may be assigned in
some cases that might otherwise be a temporary bare area.
**Water**
In the PFT$_{local}$ product, the per-pixel fraction of surface water PFT is estimated for pixels of all classes except the permanent
snow and ice class (**Table 1**). The PFT$_{local}$ product indicates around 142,000 km$^2$ of water cover globally among pixels of all
classes except the water body class (code 210). Only two classes – a sparse vegetation sub-class (code 151) and a needleleaved
deciduous tree cover sub-class (code 82) – have no pixels with inland water cover (Table 2), but both classes have extremely
limited total areal coverage, each accounting for only a few square kilometres of area globally. Classes with significant water
coverage include: needleleaved evergreen tree cover classes 70 and 71 (40,000 km$^2$ combined); sparse vegetation class 150
(20,000 km$^2$); lichens and mosses class 140 (14,000 km$^2$); flooded shrub/herbaceous cover class 180 (12,000 km$^2$); and bare
area class 200 (12,000 km$^2$). Coverage of water PFT in pixels of the non-water body classes is especially prevalent in the
boreal region. Classes with the highest fractional composition of inland water – calculated as the area-weighted mean among
all pixels of the class globally (Table 3) – include the flooded tree cover class 170 (2.3 %), the needleleaved evergreen tree
cover class 72 (1.3 %), and the lichens and mosses class 140 (0.9 %).

The PFT$_{local}$ product indicates 3 % (91,000 km$^2$) lower inland water fractional cover than the PFT$_{global}$ product distribution.
While the non-water body classes have a total inland water PFT cover of 142,000 km$^2$ in the PFT$_{local}$ product (compared to 0
km$^2$ from the PFT$_{global}$), the PFT$_{local}$ product indicates a lower inland water PFT area in the water body class than does the CWT
(difference of 233,000 km$^2$). The difference in the water body class occurs because the PFT$_{local}$ product allows up to 14 %
vegetation cover in this class whereas the CWT assumes 100 % water PFT. PFTs with significant global coverage in water
body class pixels in the PFT$_{local}$ product include natural grasses (183,000 km$^2$) and needleleaved evergreen trees (31,000 km$^2$)
with smaller contributions from the other tree types.



**Bare**

In the PFT$_{local}$ product, bare soil PFT occurs in the bare area classes (codes 200–202) and the sparse vegetation classes (codes 150–153), accounting for 19.4 million km$^2$ and 7.6 million km$^2$ bare soil area, respectively, at the global scale (Table 2). The global area-weighted mean bare soil percentages are 85.9 % in sparse vegetation class pixels and 99.9 % in bare area class pixels, which are nearly identical to the compositions suggested by the global CWT (85 % for sparse vegetation classes and 100 % for bare area classes). Cumulatively for these classes, the PFT$_{local}$ product suggests only 0.2 % lower bare soil PFT coverage at the global scale relative to the assumed distribution in the PFT$_{global}$ dataset (difference of 65,000 km$^2$). In the PFT$_{local}$ product, the bare area classes contain, in addition to bare soil PFT, inland water PFT (13,000 km$^2$) and tree cover (1,000 km$^2$).

The PFT $_{local}$ product does not include bare soil PFT in the shrubland classes, while the PFT$_{global}$ dataset assumes 20 % bare soil for the non-flooded shrubland classes 120–122. Because the non-flooded shrubland class pixels have such a large extent globally (13.3 million km$^2$), the PFT$_{global}$ dataset suggests 2.7 million km$^2$ of additional bare soil in such pixels relative to the PFT product. Differences in the distribution of bare area between the PFT $_{local}$ product and the PFT$_{global}$ are especially pronounced in the U.S. intermountain west, parts of southern and eastern Africa, the northern coast of Australia, and the highlands of Argentina and Brazil, as these are regions with significant shrubland class cover. In the PFT $_{local}$ product, all residual area in the shrubland class pixels that is not assigned as surface water or woody vegetation (trees and/or shrubs) based on the auxiliary input data is assigned as natural grass cover rather than bare soil. Since the PFT $_{local}$ product is built mainly for application to land surface models, the actual presence of grass vegetation vs. bare soil will be determined by the model given simulated or prescribed local climate conditions.

**Built fraction**

Both the PFT$_{local}$ product and the PFT$_{global}$ assign built PFT only to pixels of the urban class (code 190). The presence of a built PFT is not universal in land surface or Earth system models; for example, the current version of the ORCHIDEE land-surface model considers built areas to be 80 % bare soil and 20 % grasslands. The cross-walking of land cover classes to PFTs for the urban class strongly depends on the framework used to calculate surface fluxes in the urban environment and therefore inter-model variation in the global CWT may be stronger for the urban class than for vegetated classes. The global CWT used for this analysis assigns 100 % of the urban class as built PFT. For comparison, the JULES land surface model assigns urban class pixels as 75 % built and 25 % bare soil.

The PFT$_{local}$ product suggests 477,000 km$^2$ of built area globally (Table 2), which corresponds to an area-weighted mean composition of 73.7 % built PFT in urban class pixels (Table 3). The auxiliary inputs suggest that about 1.6 % of urban class pixels have 0 % built PFT coverage. This suggests a mismatch between the land cover classification and the auxiliary inputs for a small number of pixels, which could be related to a mismatch in the time stamp of the auxiliary inputs (2014) relative to the land cover dataset. Considering all urban class pixels, 6.2 % have built PFT of 0–25 %, 7.8 % have built PFT of 26–50 %, 31.9 % have built PFT of 51–75 %, and 54.1 % have built PFT of 76–100 %. As area-weighted means, the non-built portion of urban class pixels is 25.1 % natural grass cover, 0.3 % inland water, and 0.9 % tree cover. The increased spatial heterogeneity in urban class pixels due to the PFT$_{local}$ product is readily apparent in Figure 2, which shows the PFT distribution for Amsterdam, the Netherlands. The more realistic characterization of the urban environment in the PFT$_{local}$ product that gives more variability of built PFT coverage within a city should allow a more faithful representation of urban surface fluxes in land-surface models.

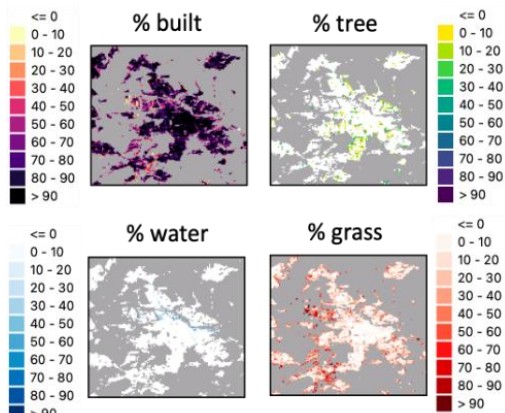

599

**Figure 2.** Percentage cover in 2010 for built, total tree, grass, and inland water PFTs in Amsterdam, the Netherlands, in the PFT$_{local}$ product.

**Permanent snow and ice**

The permanent snow and ice in PFT$_{local}$ accounts for 14.7 million km$^2$ of area globally, largely in Greenland and Antarctica, but also in the Arctic and mountainous regions of Asia. The PFT$_{local}$ product and PFT$_{global}$ dataset indicate identical coverage for this PFT since both datasets assign 100 % snow and ice PFT to the permanent snow and ice class (code 220) and 0 % snow and ice PFT to all other classes.

607    **4 Modelling results**

608    **4.1 ORCHIDEE simulations: new PFT product (PFT$_{local}$) vs PFT maps based on global CWT (PFT$_{global}$)**

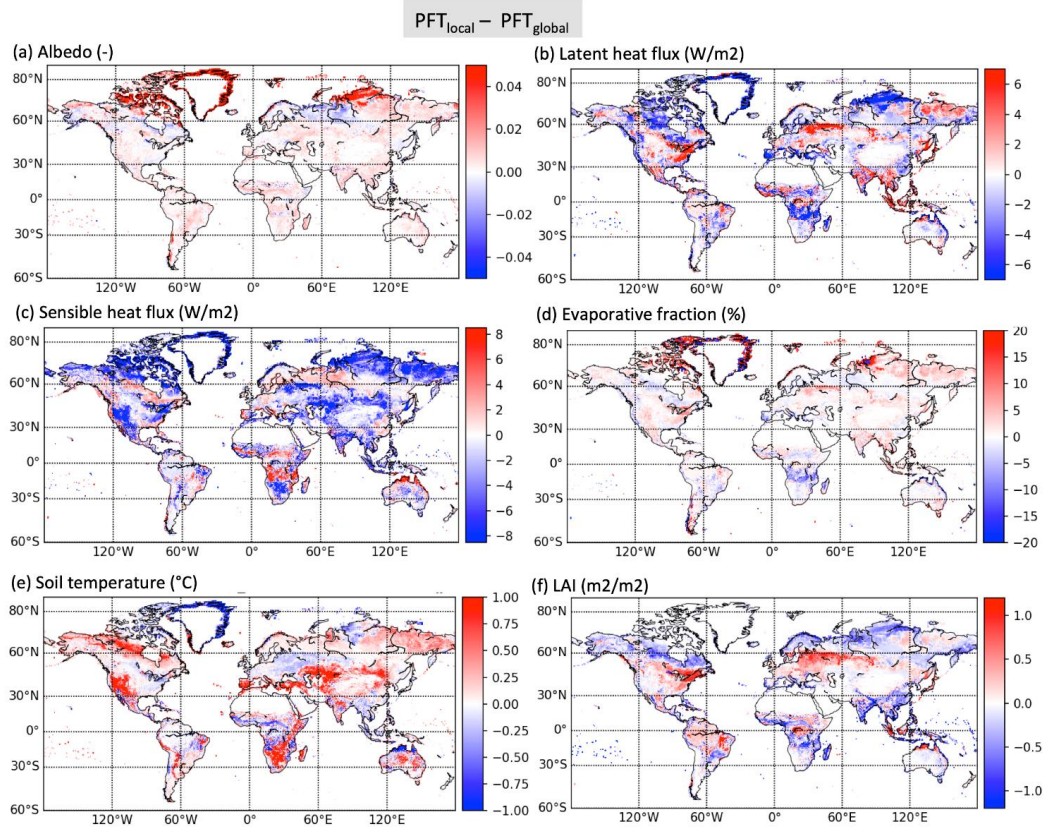

**Figure 3.** Differences in (a) albedo, (b) latent heat flux, (c) sensible heat flux, (d) evaporative fraction (Latent heat flux / (Latent + Sensible heat fluxes), (e) soil surface temperature, and (f) Leaf Area Index (LAI) simulated by the ORCHIDEE model between the new PFT (PFT$_{local}$) and the old PFT distributions (PFT$_{global}$), for the summer (June - July - August, northern hemisphere) of year 2010.

In this section, we compare the results of two ORCHIDEE simulations performed, respectively, by applying the old standard
PFT maps (PFT$_{global}$) and the new PFT product derived in this study (PFT$_{local}$). The results are shown for the year 2010.
The impacts of the changes in the land surface representation between the local and global PFT maps on the surface albedo,
latent and sensible heat fluxes, evaporative fraction (ratio of latent heat flux to the sum of latent and sensible heat fluxes),
surface temperature, and the LAI are shown in Figure 3. Averaged differences (local vs global) for the northern hemisphere
summer period (June-July-August, JJA) were plotted here to highlight the main changes but the plots at the annual scale are
also given in the Supplementary Information. The results show that the energy, water, and carbon fluxes are mainly (and
significantly) impacted in the regions where woody vegetation was replaced by grasslands or where the bare soil fraction has
changed. Since, in ORCHIDEE, the shrub PFTs are assigned to tree PFTs, the regions highlighted in Sect. 3 with significant
fractions of shrub losses or gains in profit of grasses show the largest changes. Given that tree PFTs present a lower albedo,
higher roughness (linked to vegetation height) and maximum transpiration capacity, and higher LAI and biomass, the simulated
differences between the two simulations show coherent features across the different variables. In summer, surface albedo
increases up to 4 % (absolute deviation) in the northern boreal regions because of the decrease of shrubs and the increase of
grasslands and in some regions (like in the Taymir peninsula) the increase of bare soils. More southern of this boreal zone,
both in Eurasia and North America, the increase of trees and decrease of shrubs, show opposite variations. In the tropical



region (between 0° and 30°S), the PFT changes principally concern differences in the shrubs/grasses partition at the benefit of
grasslands. In these regions, the tree fraction decrease results in a slight increase of the albedo around 2 % (absolute deviation).
At the annual scale (Figure B1), the larger impact of the PFT differences in the high latitudes is explained by the cumulative
impact of changes in snow cover. Indeed, snow melting is more rapid on tree cover compared to grasslands, inducing a shorter
duration of the snow cover with high albedo values, leading to even more differences between short and high vegetation albedo
values.

Surface albedo differences (impacting surface net radiation) combined with roughness changes (impacting turbulent
exchanges) explain generally the surface flux variations. The balance between the two effects varies according to the latitude
following the amount of solar radiation: in the northern latitudes, the impact of surface roughness is larger than in more
southern ones. In the tropics, we observe a decrease in the turbulent fluxes where the albedo is larger, explaining the lower
evapotranspiration and lower GPP, with different partitions when comparing arid and humid zones. For example, the
consequences of a decrease of shrubs to the benefit of grasses do not have the same effects on the heat flux partition according
to the water availability. In regions where soil moisture limits evapotranspiration, like central Africa (south of the Democratic
Republic of the Congo) or the Sahel, fewer trees lead to less evapotranspiration up to 6 $Wm^{-2}$ in annual mean, and larger
sensible heat flux at the same level, whereas in the northern latitudes like in eastern Siberia, fewer shrubs lead to larger
evapotranspiration and lower sensible heat flux. This is summarized in the representation of the evaporative fraction which
shows opposite variations in these regions.

The surface temperature, as the result of the energy and water budgets, shows differences in line with the sensible heat flux
variations, with larger temperatures where the sensible heat flux has decreased. The differences in summer and in annual mean
are significant and can reach 1 K but can show differences up to 3 K on a daily scale.

LAI differences are in coherence with the PFT differences: lower values where woody vegetation was replaced by grasses,
except in eastern Siberia and northern Australia where the increase of net radiation favored transpiration and GPP and, finally,
LAI. The LAI variations may reach 1 $m^2m^{-2}$ in some regions like southeastern Canada or Central Europe, where the broadleaf
deciduous trees have increased in the PFT$_{local}$ map.

Figure 4 illustrates the impacts on the above-ground biomass (AGB) with the tree cover variations. To see if the biomass
changes are more realistic, they have been compared to the ESA CCI Biomass product, version 3 (ESA$_{CCI}$ Biomass, Santoro
and Cartus, 2019; Santoro et al., 2021) aggregated at 0.5° resolution. Note that, unlike for the turbulent fluxes discussed above,
the change in AGB between low and high vegetation covers should be large enough and thus easier to evaluate. In Figure 4ab,
we first compare the simulated AGB with the new PFTs (PFT$_{local}$) to the ESA$_{CCI}$ Biomass product, which highlights some
issues related to ORCHIDEE model deficiencies and also, in part, to relatively large errors in the ESA$_{CCI}$ Biomass product,
especially for high AGB. -The model simulates too low AGB on average with a large underestimation over the tropical forests,
which cannot be due to the PFT cover (above 90 % forest cover). Over temperate and high latitudes, we also find significant
model AGB underestimation. The improvements/degradations with respect to changing the PFT distribution (Figure 4d; where
the mean errors between the two simulations performed with PFT$_{local}$ and PFT$_{global}$ are represented), provide contrasting results
between regions. The benefits of the new PFT$_{local}$ maps (blue color in Figure 4d) are visible in northeast Europe, the eastern
USA and in Democratic Republic of the Congo where the increase of tree fraction (Figure 4c) and biomass seems to be in
better agreement with the remote sensing AGB product. In the other regions, where the tree fractions decreased (northern
Canada and Europe, Sahel, Angola, Zambia and southern China, Figure 4c), the associated decrease of biomass leads to larger
errors compared to the AGB satellite product. In the western USA (California), the losses of tree PFTs to the benefit of



grasslands did not impact the simulated biomass since, in these arid regions, the trees have very low productivity comparable
to grasses and thus similar low biomass values (less than 1 KgCm⁻²).

Overall, these results highlight the importance and impact of land surface PFT distribution on simulated energy, water, and
carbon fluxes as well as carbon stocks in global land surface models.

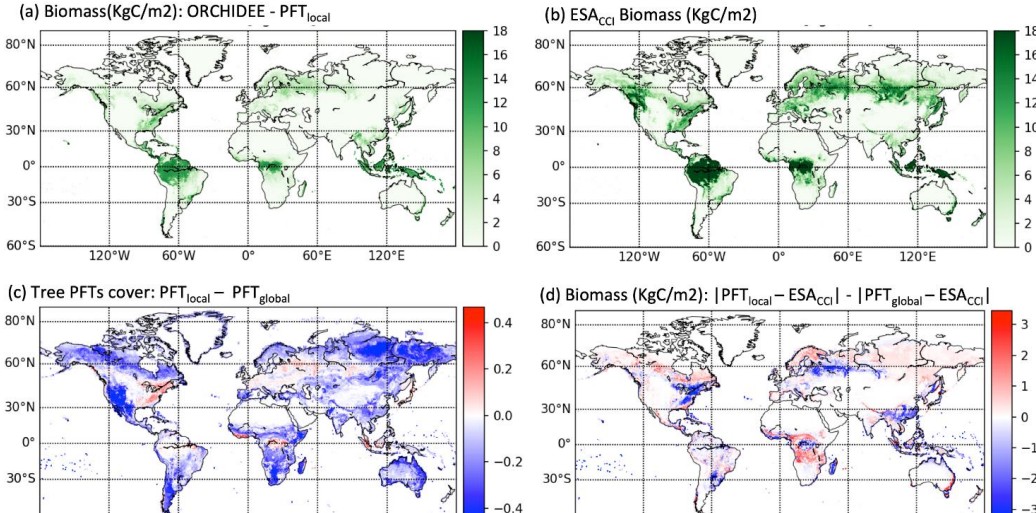


**Figure 4.** Above Ground Biomass (AGB) (a) simulated by the ORCHIDEE model with the PFT$_{local}$ dataset and (b) observed by the ESA$_{CCI}$
Biomass product version v3, for the year 2010 (Santoro and Cartus (2019)); (c) Differences in the tree PFT fraction prescribed; (d) Difference
between the mean bias of simulated versus ESA$_{CCI}$ Biomass AGB between the new (PFT$_{local}$) and the former (PFT$_{global}$) distributions of
PFTs. Negative values indicate a decrease in the bias from the PFT$_{global}$ to the PFT$_{local}$.
**4.2 Evaluation of DGVM (JULES-TRIFFID) using PFT fractions**
The impact of using the new PFT distributions (PFT$_{local}$) as a benchmark for JULES-TRIFFID dynamic vegetation is shown
in Figure 5. In contrast to results shown in Sect. 4.1, differences found here indicate the value of the new PFT distributions as
a product for model evaluation, rather than a direct improvement of model predictions. When compared to PFT$_{global}$ ('CWT'),
JULES-TRIFFID indicates significant over-estimation of tree cover in tropical savannahs, and under-estimation of tree cover
in boreal northeast Russia. Additionally, comparison with the global CWT product (PFT$_{global}$) indicates that JULES-TRIFFID
under-estimates shrub cover in tropical savannahs in South America, Africa, and Australia, as well as many semi-arid regions
such as western North America. Biases in grass cover are more spatially heterogeneous, but comparison with the global CWT
indicates that JULES-TRIFFID strongly over-estimates in northeast Russia and northern Australia.

When using the new PFT distributions as a benchmark, many of these biases are reduced, as indicated by green areas in column
"c" of Figure 5. In particular, northeast boreal Russia shows reduced biases in tree, shrub, and grass cover. Globally, using the
new PFT distributions results in a reduction in biases in shrub cover in JULES-TRIFFID in almost every part of the world,
particularly savannahs and semi-arid regions (Figure 4d). Whilst no large areas showed a large increase in bias, some areas
did show increases in bias of up to 25 %, such as tropical forests (10 % increase), grass cover in tropical savannahs (15 to 25
%) and northern high latitudes (10 to 20 %), and bare cover in arid regions (up to 10 % increase).

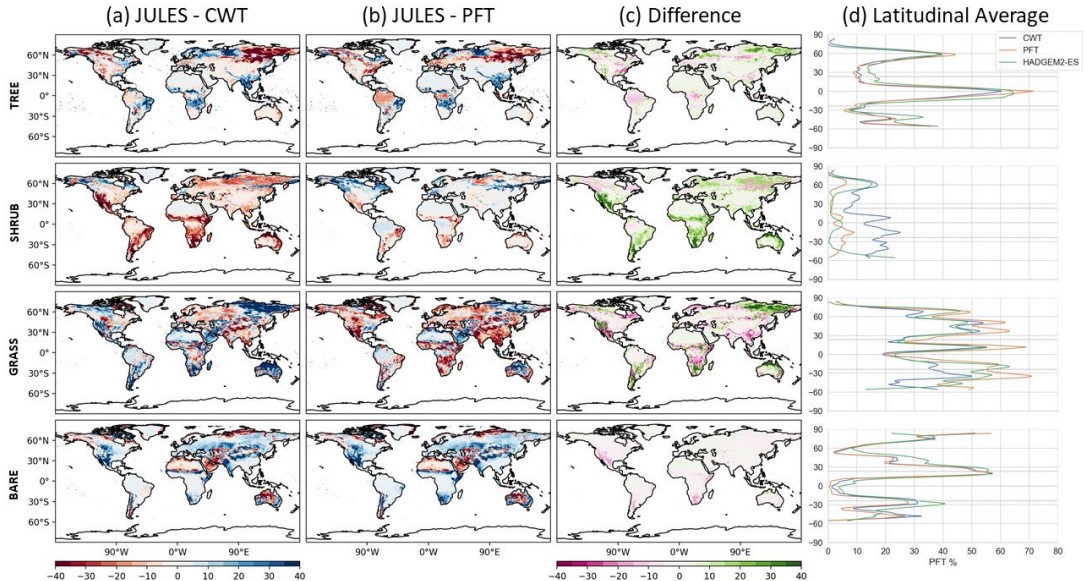


**Figure 5.** Comparison of JULES PFT distributions to both the CWT ($PFT_{global}$) and PFT ($PFT_{local}$) products for major vegetation types. Rows show each major surface type (Tree, Shrub, Grass, Bare), whilst rows show (a) JULES vegetation distribution compared to the global CWT; (b) the same compared to new $PFTl_{local}$ distributions; (c) the difference between (a) and (b), where green (pink) indicates positive or negative anomalies evaluate closer to 0 (further away from 0) using new $PFT_{local}$ distributions; (d) absolute latitudinal average fractions for each major vegetation type from CWT, PFT, and JULES.

## 5 Conclusion and perspectives

The new PFT product ($PFT_{local}$) was generated to reduce the cross-walking component of uncertainty by adding spatial variability to the PFT composition within a LC class. This work moved beyond fine-tuning the cross-walking approach for specific LC classes or regions and, instead, separately quantifies the PFT fractional composition for each 300 m pixel globally. The result is a dataset representing the cover fractions of 14 PFTs at 300 m for each year in the 1992–2020 era, consistent with the CCI MRLC map for the corresponding year. The $PFT_{local}$ dataset exhibits intraclass spatial variability in PFT fractional cover at the 300 m pixel level and is complementary to the CCI medium resolution multi-mission LC map series since the derived PFT fractions maintain consistency with the original LC class legend.

The $PFT_{local}$ dataset provides a more faithful representation of PFT distributions because it draws on high-resolution peer-reviewed mapping of specific vegetation classes to refine global assumptions about PFT fractions. In many cases, the global CWT presented a reasonable approximation for estimating PFT fractions within many land cover classes as shown by the fractions estimated from the auxiliary products falling close to that suggested by the global CWT.

Note that a recent study by Marie et al. (2022) followed the same objective of refining the global CWT (used to map the ESA land cover classes onto PFTs) but with a different approach. Instead of using the tree cover dataset from Hansen et al. (2013), they valorised a map of above-ground biomass over Africa (Bouvet et al., 2018) to define local CWTs, using the information from AGB to better constrain the partition between tree and short vegetation PFTs, for each LC class. As shown in our study, they found that LC class 10 (rainfed cropland) in the Sahel should contain tree PFTs which correspond to tree crops (Figure A1). Overall, these efforts highlight the benefits of using additional high-resolution products, like tree cover, AGB, etc. when translating land cover into PFT distributions for land surface models. Merging all sources of information into a coherent PFT product remained however a difficult task. This study demonstrated that using the consistent CCI MRLC time series and



maintaining deference to the original LCCS class in the combination rules allowed bringing these auxiliary data into
consistency.

Changing the PFT distribution in the ORCHIDEE model (PFT$_{local}$ vs PFT$_{global}$) induces significant impacts on the simulated
water, energy, and carbon fluxes as well as on the modelled carbon stocks. These differences are coherent with changes in
surface properties (albedo, roughness, type of cover) induced by changes is PFT types (mainly tree vs short vegetation and
bare soil covers). However, it is not possible and beyond the scope of the paper to evaluate globally and quantitatively model
improvements due to changes in PFTs given i) existing model biases that have been partly compensated by previous model
parameters tuning with the old PFT maps (PFT$_{global}$) and ii) the large uncertainty still associated with data-driven products at
global scale. We initiated an evaluation with AGB; however, the new simulated biomass (induced by PFT changes) is not
always closer to the satellite ESA$_{CCI}$ AGB product. In addition, the fact that ORCHIDEE does not differentiate shrubs and
trees limits such biomass evaluation. Additional simulations/tests with more models and a more comprehensive evaluation
with a larger ensemble of variables and data-driven products are therefore needed to quantify the benefits of the PFT$_{local}$ maps.

Using the PFT$_{local}$ as a benchmark improves the evaluation of every major surface type in the JULES-TRIFFID dynamic
vegetation model, particularly shrub cover. This allows a new perspective on priorities for dynamic vegetation model
development.

The user tool described in Poulter et al. (2015) has been reformatted such that it can be applied directly to the new PFT map
series to create user-specific ready-to-use inputs for LSMs. The user tool creates model-ready inputs at user specification,
which greatly expands the ease of use of the product both within and beyond the modelling community. The PFT dataset is
designed primarily for use in land surface and Earth system models. For the vegetated classes except for sparse vegetation, the
entire non-water fraction of the 300 m pixel is assigned as vegetation, allowing the actual presence of grass vegetation to be
determined by the land surface models. For use outside of modelling, this could introduce some bias (e.g., underestimating
bare soil cover in some pixels and overestimating grass cover), but the fractions of the high biomass veg types (trees and
shrubs) can be used for non-modelling use cases.

Production of the PFT$_{local}$ product is dependent on the availability and quality of the auxiliary datasets at a spatial resolution
higher than 300 m; this is especially critical for mapping the shrubland class. With the combined information of the
phenological attribute of the ESA CCI LC classes, the percentage of tree canopy cover from Hansen et al. (2013), and the
GEDI product (Potapov et al., 2021), it was possible, for the first time, to map four shrubland classes at the global scale:
broadleaved evergreen, broadleaved deciduous, needleleaved evergreen, and needleleaved deciduous. Yet, further research is
still needed to improve the estimation of shrubland class pixels north of 52°N (i.e., outside of the extent of the GEDI product).
The urban PFT would benefit from separating impervious surfaces from buildings. Finnally, the current workflow should
further be tested against annual ancillary product updates as operational production of very-high-resolution datasets becomes
the norm. The proposed methodology is automated so that the PFT dataset will be updated annually as new annual land cover
maps are produced in C3S.



**Tables**

**Table 1.** For each of the 22 global and 15 regional land cover classes of the CCI MRLC map series, listed are the set of contributing PFTs with the possibility for non-zero fractional cover. The regional land cover classes with codes ending with 1, 2, or 3, are thematically richer than the global classes but can be found only at the regional scale depending on training data availability.

| Class code | Class description | PFTs for which non-zero fractions are permitted in the PFT product |
|---|---|---|
| 10 | Rainfed cropland | Trees, water, managed grass |
| 11 | Rainfed cropland – herbaceous cover | Trees, water, managed grass |
| 12 | Rainfed cropland – tree or shrub cover | Trees, water, managed grass |
| 20 | Irrigated or post-flooding cropland | Trees, water, managed grass |
| 30 | Mosaic: > 50 % cropland/ < 50 % natural tree, shrub, herbaceous cover | Trees, water, managed grass |
| 40 | Mosaic: > 50 % natural tree, shrub, herbaceous cover/ < 50 % cropland | Trees, water, natural grass, managed grass |
| 50 | > 15 % broadleaved evergreen tree cover | Broadleaved evergreen trees, water, natural grass |
| 60 | > 15 % broadleaved deciduous tree cover | Broadleaved deciduous trees, water, natural grass |
| 61 | > 40 % broadleaved deciduous tree cover | Broadleaved deciduous trees, water, natural grass |
| 62 | 15–40 % broadleaved deciduous tree cover | Broadleaved deciduous trees, water, natural grass |
| 70 | > 15 % needleleaved evergreen tree cover | Needleleaved evergreen trees, water, natural grass |
| 71 | > 40 % needleleaved evergreen tree cover | Needleleaved evergreen trees, water, natural grass |
| 72 | 15–40 % needleleaved evergreen tree cover | Needleleaved evergreen trees, water, natural grass |
| 80 | > 15 % needleleaved deciduous tree cover | Needleleaved deciduous trees, water, natural grass |
| 81 | > 40 % needleleaved deciduous tree cover | Needleleaved deciduous trees, water, natural grass |
| 82 | 15–40 % needleleaved deciduous tree cover | Needleleaved deciduous trees, water, natural grass |
| 90 | Mixed leaf type (broadleaved and needleleaved) tree cover | Trees, water, natural grass |
| 100 | Mosaic: > 50 % tree and shrub cover / < 50 % herbaceous cover | Trees, water, natural grass |
| 110 | Mosaic: > 50 % herbaceous cover / < 50 % tree and shrub cover | Trees, water, natural grass |
| 120 | Shrubland | Trees, water, natural grass, shrubs |
| 121 | Evergreen shrubland | Trees, water, natural grass, shrubs |
| 122 | Deciduous shrubland | Trees, water, natural grass, shrubs |
| 130 | Grassland | Trees, water, natural grass |
| 140 | Lichens and mosses | Water, natural grass |
| 150 | Sparse vegetation: < 15 % tree, shrub, herbaceous cover | Trees, water, natural grass, bare soil |
| 151 | Sparse vegetation: < 15 % tree cover | Trees, water, natural grass, bare soil |
| 152 | Sparse vegetation: < 15 % shrub cover | Trees, water, natural grass, bare soil |
| 153 | Sparse vegetation: < 15 % herbaceous cover | Trees, water, natural grass, bare soil |
| 160 | Flooded tree cover – fresh or brackish water | Trees, water, natural grass |
| 170 | Flooded tree cover – saline water | Trees, water, natural grass |
| 180 | Flooded shrub or herbaceous cover – fresh, saline, or brackish water | Trees, water, natural grass, shrubs |
| 190 | Urban areas | Trees, water, natural grass, built |
| 200 | Bare areas | Trees, water, bare soil |
| 201 | Consolidated bare areas | Trees, water, bare soil |
| 202 | Unconsolidated bare areas | Trees, water, bare soil |
| 210 | Water body | Trees, water, natural grass |
| 220 | Permanent snow and ice | Snow and ice |






**Table 2.** Global areal cover (1000 km²) of each PFT by land cover class for 2010 in the PFT$_{local}$ product.

| Class | Bare soil | Built | Managed grasses | Natural grasses | Snow/ice | Water[1] | BD trees | BE trees | ND trees | NE trees | BD shrubs | BE shrubs | ND shrubs | NE shrubs |
|---|---|---|---|---|---|---|---|---|---|---|---|---|---|---|
| 10 | 0 | 0 | 7729.7 | 0 | 0 | 2.3 | 175.1 | 199.5 | 0.7 | 36 | 0 | 0 | 0 | 0 |
| 11 | 0 | 0 | 6774.9 | 0 | 0 | 1.5 | 110.4 | 112.9 | 4.1 | 19.7 | 0 | 0 | 0 | 0 |
| 12 | 0 | 0 | 155.1 | 0 | 0 | 0.1 | 4.6 | 29.8 | 0 | 0.6 | 0 | 0 | 0 | 0 |
| 20 | 0 | 0 | 2415.5 | 0 | 0 | 1.2 | 19 | 7.4 | 0.2 | 1.8 | 0 | 0 | 0 | 0 |
| 30 | 0 | 0 | 2803 | 0 | 0 | 1.1 | 123.2 | 467.2 | 0.8 | 39 | 0 | 0 | 0 | 0 |
| 40 | 0 | 0 | 1557.4 | 1247.9 | 0 | 1.2 | 195.1 | 493.2 | 4.7 | 65.1 | 0 | 0 | 0 | 0 |
| 50 | 0 | 0 | 0 | 1262.6 | 0 | 4.3 | 0 | 11476.1 | 0 | 0 | 0 | 0 | 0 | 0 |
| 60 | 0 | 0 | 0 | 2237.9 | 0 | 1.7 | 3599.6 | 0 | 0 | 0 | 0 | 0 | 0 | 0 |
| 61 | 0 | 0 | 0 | 337 | 0 | 0.2 | 538.6 | 0 | 0 | 0 | 0 | 0 | 0 | 0 |
| 62 | 0 | 0 | 0 | 2673.9 | 0 | 0.2 | 1000 | 0 | 0 | 0 | 0 | 0 | 0 | 0 |
| 70 | 0 | 0 | 0 | 2411.4 | 0 | 21.9 | 0 | 0 | 0 | 4060.4 | 0 | 0 | 0 | 0 |
| 71 | 0 | 0 | 0 | 710.7 | 0 | 18 | 0 | 0 | 0 | 1720.3 | 0 | 0 | 0 | 0 |
| 72 | 0 | 0 | 0 | 0.7 | 0 | 0 | 0 | 0 | 0 | 0.3 | 0 | 0 | 0 | 0 |
| 80 | 0 | 0 | 0 | 2977.6 | 0 | 4 | 0.1 | 0 | 2143.5 | 0 | 0 | 0 | 0 | 0 |
| 81 | 0 | 0 | 0 | 0.7 | 0 | 0 | 0 | 0 | 4.1 | 0 | 0 | 0 | 0 | 0 |
| 82 | 0 | 0 | 0 | 0 | 0 | 0 | 0 | 0 | 0 | 0 | 0 | 0 | 0 | 0 |
| 90 | 0 | 0 | 0 | 441 | 0 | 1.6 | 674.1 | 63.7 | 77.8 | 918.5 | 0 | 0 | 0 | 0 |
| 100 | 0 | 0 | 0 | 2443.7 | 0 | 2.6 | 233.8 | 329.3 | 43.4 | 354 | 0 | 0 | 0 | 0 |
| 110 | 0 | 0 | 0 | 977.3 | 0 | 0.5 | 66.1 | 26.1 | 3.5 | 11.1 | 0 | 0 | 0 | 0 |
| 120 | 0 | 0 | 0 | 7246.3 | 0 | 4 | 176.3 | 125.5 | 3.9 | 80.7 | 1746.8 | 632.9 | 164.7 | 724.5 |
| 121 | 0 | 0 | 0 | 142.5 | 0 | 0 | 1.6 | 26.3 | 5.3 | 0.6 | 2.7 | 31.2 | 26.8 | 1.5 |
| 122 | 0 | 0 | 0 | 1294.6 | 0 | 1.1 | 41 | 68.5 | 10.4 | 4.2 | 211 | 154.6 | 293.8 | 86.8 |
| 130 | 0 | 0 | 0 | 13338.8 | 0 | 5.8 | 144 | 159 | 8.5 | 47.4 | 0 | 0 | 0 | 0 |
| 140 | 0 | 0 | 0 | 1476.9 | 0 | 14.2 | 0 | 0 | 0 | 0 | 0 | 0 | 0 | 0 |
| 150 | 7254.8 | 0 | 0 | 1157.7 | 0 | 20.4 | 0.5 | 1.3 | 0.6 | 12 | 0 | 0 | 0 | 0 |
| 151 | 0 | 0 | 0 | 0 | 0 | 0 | 0 | 0 | 0 | 0 | 0 | 0 | 0 | 0 |
| 152 | 63 | 0 | 0 | 8.9 | 0 | 0.2 | 0.1 | 0 | 0.2 | 1.1 | 0 | 0 | 0 | 0 |
| 153 | 323.8 | 0 | 0 | 52.7 | 0 | 0.1 | 0 | 0 | 0 | 0 | 0 | 0 | 0 | 0 |
| 160 | 0 | 0 | 0 | 200.4 | 0 | 2.3 | 71.7 | 442.6 | 27.4 | 151.4 | 0 | 0 | 0 | 0 |
| 170 | 0 | 0 | 0 | 86.1 | 0 | 5 | 12.3 | 110 | 4.6 | 0.8 | 0 | 0 | 0 | 0 |
| 180 | 0 | 0 | 0 | 1231.9 | 0 | 11.7 | 12.7 | 15.1 | 5.5 | 51.8 | 122.3 | 56.6 | 47.7 | 362.7 |
| 190 | 0 | 476.7 | 0 | 162.5 | 0 | 1.9 | 2.5 | 1.3 | 0.2 | 2.1 | 0 | 0 | 0 | 0 |
| 200 | 19156.9 | 0 | 0 | 0 | 0 | 12.2 | 0.4 | 0 | 0.2 | 0.5 | 0 | 0 | 0 | 0 |
| 201 | 108.8 | 0 | 0 | 0 | 0 | 0.3 | 0 | 0 | 0 | 0 | 0 | 0 | 0 | 0 |
| 202 | 97.2 | 0 | 0 | 0 | 0 | 0.1 | 0 | 0 | 0 | 0 | 0 | 0 | 0 | 0 |
| 210 | 0 | 0 | 0 | 182.6 | 0 | 365991.8 | 7.2 | 8.9 | 3.1 | 31.3 | 0 | 0 | 0 | 0 |
| 220 | 0 | 0 | 0 | 0 | 14694.2 | 0 | 0 | 0 | 0 | 0 | 0 | 0 | 0 | 0 |


**Table 3.** Percentage PFT composition by class for 2010 calculated as an area-weighted mean over all pixels of the class globally.

| Class code | Bare soil | Built | Managed grasses | Natural grasses | Snow/ice | Water[2] | BD trees | BE trees | ND trees | NE trees | BD shrubs | BE shrubs | ND shrubs | NE shrubs |
|---|---|---|---|---|---|---|---|---|---|---|---|---|---|---|
| 10 | 0.0 | 0.0 | 94.9 | 0.0 | 0.0 | 0.0 | 2.1 | 2.5 | 0.0 | 0.4 | 0.0 | 0.0 | 0 | 0.0 |
| 11 | 0.0 | 0.0 | 96.5 | 0.0 | 0.0 | 0.0 | 1.6 | 1.6 | 0.1 | 0.3 | 0.0 | 0.0 | 0 | 0.0 |
| 12 | 0.0 | 0.0 | 81.6 | 0.0 | 0.0 | 0.0 | 2.4 | 15.7 | 0.0 | 0.3 | 0.0 | 0.0 | 0 | 0.0 |
| 20 | 0.0 | 0.0 | 98.8 | 0.0 | 0.0 | 0.1 | 0.8 | 0.3 | 0.0 | 0.1 | 0.0 | 0.0 | 0 | 0.0 |
| 30 | 0.0 | 0.0 | 81.6 | 0.0 | 0.0 | 0.0 | 3.6 | 13.6 | 0.0 | 1.1 | 0.0 | 0.0 | 0 | 0.0 |
| 40 | 0.0 | 0.0 | 43.7 | 35.0 | 0.0 | 0.0 | 5.5 | 13.8 | 0.1 | 1.8 | 0.0 | 0.0 | 0 | 0.0 |
| 50 | 0.0 | 0.0 | 0.0 | 9.9 | 0.0 | 0.0 | 0.0 | 90.1 | 0.0 | 0.0 | 0.0 | 0.0 | 0 | 0.0 |
| 60 | 0.0 | 0.0 | 0.0 | 38.3 | 0.0 | 0.0 | 61.6 | 0.0 | 0.0 | 0.0 | 0.0 | 0.0 | 0 | 0.0 |
| 61 | 0.0 | 0.0 | 0.0 | 38.5 | 0.0 | 0.0 | 61.5 | 0.0 | 0.0 | 0.0 | 0.0 | 0.0 | 0 | 0.0 |
| 62 | 0.0 | 0.0 | 0.0 | 72.8 | 0.0 | 0.0 | 27.2 | 0.0 | 0.0 | 0.0 | 0.0 | 0.0 | 0 | 0.0 |
| 70 | 0.0 | 0.0 | 0.0 | 37.1 | 0.0 | 0.3 | 0.0 | 0.0 | 0.0 | 62.5 | 0.0 | 0.0 | 0 | 0.0 |
| 71 | 0.0 | 0.0 | 0.0 | 29.0 | 0.0 | 0.7 | 0.0 | 0.0 | 0.0 | 70.2 | 0.0 | 0.0 | 0 | 0.0 |
| 72 | 0.0 | 0.0 | 0.0 | 72.6 | 0.0 | 1.3 | 0.0 | 0.0 | 0.0 | 26.1 | 0.0 | 0.0 | 0 | 0.0 |
| 80 | 0.0 | 0.0 | 0.0 | 58.1 | 0.0 | 0.1 | 0.0 | 0.0 | 41.8 | 0.0 | 0.0 | 0.0 | 0 | 0.0 |
| 81 | 0.0 | 0.0 | 0.0 | 15.4 | 0.0 | 0.5 | 0.0 | 0.0 | 84.1 | 0.0 | 0.0 | 0.0 | 0 | 0.0 |
| 82 | 0.0 | 0.0 | 0.0 | 82.9 | 0.0 | 0.0 | 0.0 | 0.0 | 17.1 | 0.0 | 0.0 | 0.0 | 0 | 0.0 |
| 90 | 0.0 | 0.0 | 0.0 | 20.3 | 0.0 | 0.1 | 31.0 | 2.9 | 3.6 | 42.2 | 0.0 | 0.0 | 0 | 0.0 |
| 100 | 0.0 | 0.0 | 0.0 | 71.7 | 0.0 | 0.1 | 6.9 | 9.7 | 1.3 | 10.4 | 0.0 | 0.0 | 0 | 0.0 |
| 110 | 0.0 | 0.0 | 0.0 | 90.1 | 0.0 | 0.0 | 6.1 | 2.4 | 0.3 | 1.0 | 0.0 | 0.0 | 0 | 0.0 |
| 120 | 0.0 | 0.0 | 0.0 | 66.4 | 0.0 | 0.0 | 1.6 | 1.2 | 0.0 | 0.7 | 16.0 | 5.8 | 1.5 | 6.6 |
| 121 | 0.0 | 0.0 | 0.0 | 59.7 | 0.0 | 0.0 | 0.7 | 11.0 | 2.2 | 0.3 | 1.1 | 13.1 | 11.3 | 0.6 |
| 122 | 0.0 | 0.0 | 0.0 | 59.8 | 0.0 | 0.1 | 1.9 | 3.2 | 0.5 | 0.2 | 9.7 | 7.1 | 13.6 | 4.0 |
| 130 | 0.0 | 0.0 | 0.0 | 97.3 | 0.0 | 0.0 | 1.1 | 1.2 | 0.1 | 0.3 | 0.0 | 0.0 | 0 | 0.0 |
| 140 | 0.0 | 0.0 | 0.0 | 99.1 | 0.0 | 0.9 | 0.0 | 0.0 | 0.0 | 0.0 | 0.0 | 0.0 | 0 | 0.0 |
| 150 | 85.9 | 0.0 | 0.0 | 13.7 | 0.0 | 0.2 | 0.0 | 0.0 | 0.0 | 0.1 | 0.0 | 0.0 | 0 | 0.0 |
| 151 | 86.0 | 0.0 | 0.0 | 14.0 | 0.0 | 0.0 | 0.0 | 0.0 | 0.0 | 0.0 | 0.0 | 0.0 | 0 | 0.0 |
| 152 | 85.8 | 0.0 | 0.0 | 12.1 | 0.0 | 0.2 | 0.1 | 0.0 | 0.3 | 1.5 | 0.0 | 0.0 | 0 | 0.0 |
| 153 | 86.0 | 0.0 | 0.0 | 14.0 | 0.0 | 0.0 | 0.0 | 0.0 | 0.0 | 0.0 | 0.0 | 0.0 | 0 | 0.0 |
| 160 | 0.0 | 0.0 | 0.0 | 22.4 | 0.0 | 0.3 | 8.0 | 49.4 | 3.1 | 16.9 | 0.0 | 0.0 | 0 | 0.0 |
| 170 | 0.0 | 0.0 | 0.0 | 39.3 | 0.0 | 2.3 | 5.6 | 50.3 | 2.1 | 0.4 | 0.0 | 0.0 | 0 | 0.0 |
| 180 | 0.0 | 0.0 | 0.0 | 64.2 | 0.0 | 0.6 | 0.7 | 0.8 | 0.3 | 2.7 | 6.4 | 3.0 | 2.5 | 18.9 |
| 190 | 0.0 | 73.7 | 0.0 | 25.1 | 0.0 | 0.3 | 0.4 | 0.2 | 0.0 | 0.3 | 0.0 | 0.0 | 0 | 0.0 |

[1] For the water body class (code 210), the water PFT area includes 2,877,500 km² of inland water. For all other classes, all water PFT area is inland water.

[2] For the water body class (code 210), the water PFT percentage includes inland water. The area-weighted mean percentage composition of inland water PFT in water body class pixels is 0.8 %. For all other classes, all water is inland water.



| 200 | 99.9 | 0.0 | 0.0 | 0.0 | 0.0 | 0.1 | 0.0 | 0.0 | 0.0 | 0.0 | 0.0 | 0.0 | 0 | 0.0 |
| 201 | 99.7 | 0.0 | 0.0 | 0.0 | 0.0 | 0.2 | 0.0 | 0.0 | 0.0 | 0.0 | 0.0 | 0.0 | 0 | 0.0 |
| 202 | 99.9 | 0.0 | 0.0 | 0.0 | 0.0 | 0.1 | 0.0 | 0.0 | 0.0 | 0.0 | 0.0 | 0.0 | 0 | 0.0 |
| 210 | 0.0 | 0.0 | 0.0 | 0.0 | 0.0 | 99.9 | 0.0 | 0.0 | 0.0 | 0.0 | 0.0 | 0.0 | | 0.0 |
| 220 | 0.0 | 0.0 | 0.0 | 0.0 | 100.0 | 0.0 | 0.0 | 0.0 | 0.0 | 0.0 | 0.0 | 0.0 | 0 | 0.0 |

**Data availability**

The CCI PFT dataset 1992-2020 is freely, permanently, and publicly available on a web viewer: http://maps.elie.ucl.ac.be/CCI/viewer/download.php (doi: 10.5285/26a0f46c95ee4c29b5c650b129aab788, Harper et al., 2022).

**Author contribution**

PD conceived the idea and supervised the research effort with contribution from PP. CL coordinated the project. KH designed the methodology with contribution from CL. KH developed the code to generate the CCI PFT dataset 2010. AH, PP, CO, VB, RSM, and SIB designed the modelling experiments and analysed the results. GK, MB, RS, and CB designed and developed the user tool. GK and RS managed and produced the CCI PFT dataset 2010 metadata. KH wrote the original draft with contributions from CL, AH, SIB, PP, and CO. All co-authors reviewed the manuscript.

**Competing interests**

"The authors declare that they have no conflict of interest."

**Acknowledgement**

This study was carried out with the support of the European Space Agency Climate Change Initiative under the contract ESA/No.4000126564 Land_Cover_cci. The ESA CCI supported the methodological development and generation of the global PFT map series 1992-2020 as well as the climate modelling analysis. We thank Olivier Arino and Fabrizio Ramoino for their long-term support in the Land_Cover_cci project. We thank Clement Albergel for his fresh look on the work done and his review. We thank Benjamin Goffart for tailoring the CCI Land Cover web interface to visualize and interact with the CCI PFT dataset 2010.

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





**Appendix A: complementary information about the CCI PFT dataset description**

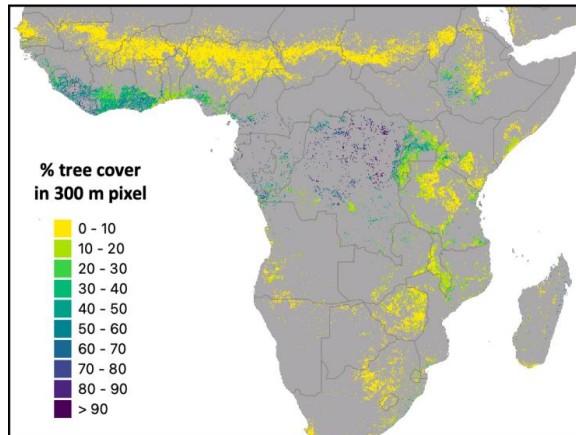

**Figure A1.** Distribution of tree cover percentage in rainfed cropland class pixels in Africa. Gray pixels belong to other classes.

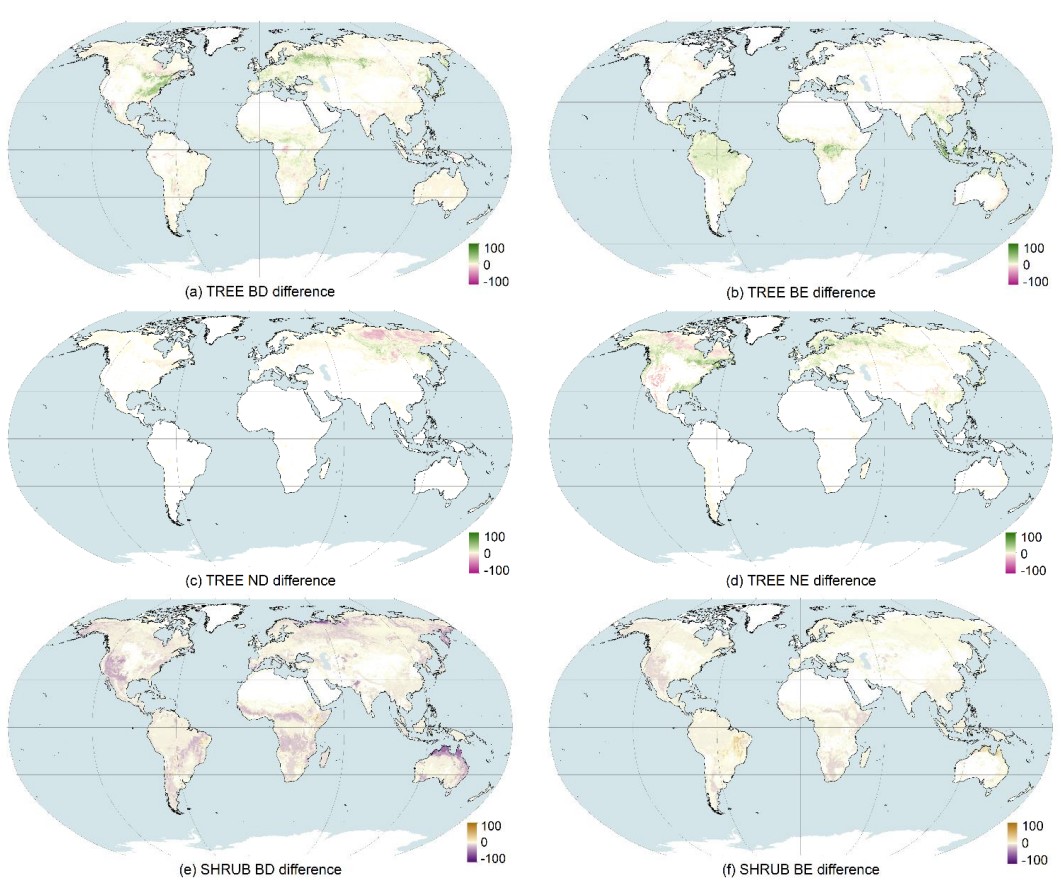

Figure A2. Absolute differences (percentage of pixel) between the 2010 PFT$_{local}$ dataset and corresponding PFT$_{global}$ maps (i.e., applying the global cross-walking scheme) for the 14 PFT types. The spatial resolution is 0.25 x 0.25 degrees.



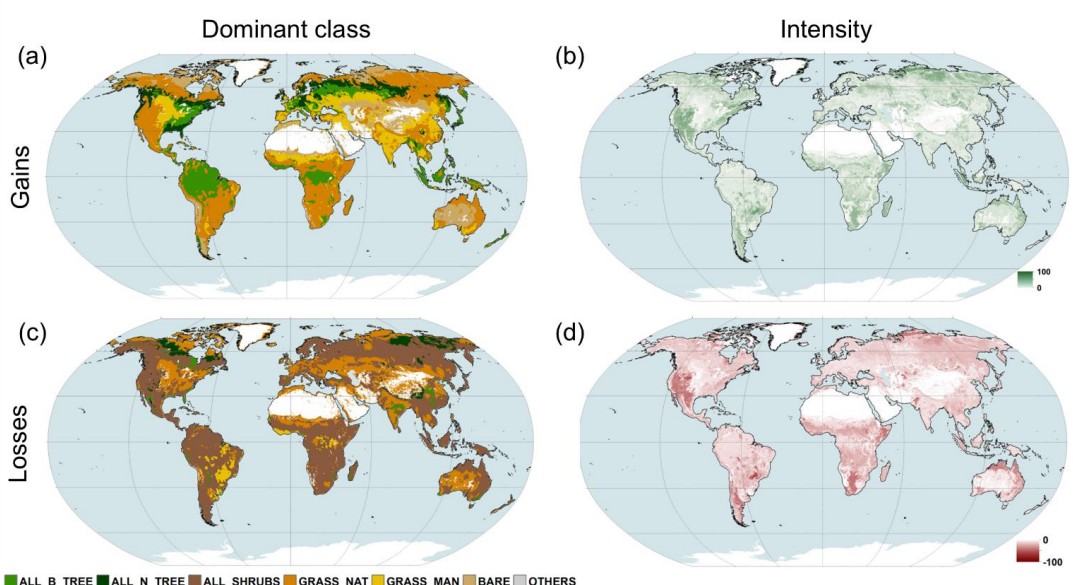

■ ALL_B_TREE ■ ALL_N_TREE ■ ALL_SHRUBS ■ GRASS_NAT ■ GRASS_MAN ■ BARE □ OTHERS
**Figure A3.** PFT with the largest increase (a) and largest loss (c) in coverage within 0.25° x 0.25° pixels in the PFT$_{local}$ dataset compared to
the PFT$_{global}$ and corresponding fractions gained (b) and lost (d). White areas remained stable in both PFT datasets.

**Table A1.** Global areal cover (1000 km²) of each PFT by land cover class for 2010 based on the most recent version of the
global CWT applied to v2.0.8 of the CCI MRLC map.

| Class | Bare soil | Built | Grass Man | Grass NAT | Snow/ice | Water[3] | BD trees | BE trees | ND trees | NE trees | BD shrubs | BE shrubs | ND shrubs | NE shrubs |
|---|---|---|---|---|---|---|---|---|---|---|---|---|---|---|
| 10 | 0 | 0 | 7328.9 | 814.3 | 0 | 0 | 0 | 0 | 0 | 0 | 0 | 0 | 0 | 0 |
| 11 | 0 | 0 | 6321.2 | 702.4 | 0 | 0 | 0 | 0 | 0 | 0 | 0 | 0 | 0 | 0 |
| 12 | 0 | 0 | 57.1 | 0 | 0 | 0 | 0 | 0 | 0 | 0 | 133.2 | 0 | 0 | 0 |
| 20 | 0 | 0 | 2200.6 | 244.5 | 0 | 0 | 0 | 0 | 0 | 0 | 0 | 0 | 0 | 0 |
| 30 | 0 | 0 | 2060.5 | 515.1 | 0 | 0 | 171.7 | 171.7 | 0 | 0 | 171.7 | 171.7 | 0 | 171.7 |
| 40 | 0 | 0 | 712.9 | 1069.4 | 0 | 0 | 267.4 | 267.4 | 0 | 0 | 534.7 | 356.5 | 0 | 356.5 |
| 50 | 0 | 0 | 0 | 0 | 0 | 0 | 0 | 11468.7 | 0 | 0 | 637.1 | 637.1 | 0 | 0 |
| 60 | 0 | 0 | 0 | 1751.8 | 0 | 0 | 2919.6 | 0 | 0 | 0 | 1167.9 | 0 | 0 | 0 |
| 61 | 0 | 0 | 0 | 131.4 | 0 | 0 | 613.1 | 0 | 0 | 0 | 131.4 | 0 | 0 | 0 |
| 62 | 0 | 0 | 0 | 1653.3 | 0 | 0 | 1102.2 | 0 | 0 | 0 | 918.5 | 0 | 0 | 0 |
| 70 | 0 | 0 | 0 | 974 | 0 | 0 | 0 | 0 | 0 | 4545.6 | 324.7 | 324.7 | 0 | 324.7 |
| 71 | 0 | 0 | 0 | 367.4 | 0 | 0 | 0 | 0 | 0 | 1714.4 | 122.5 | 122.5 | 0 | 122.5 |
| 72 | 0 | 0 | 0 | 0.5 | 0 | 0 | 0 | 0 | 0 | 0.3 | 0 | 0 | 0 | 0.3 |
| 80 | 0 | 0 | 0 | 1537.6 | 0 | 0 | 0 | 0 | 2562.6 | 0 | 128.1 | 128.1 | 640.7 | 128.1 |
| 81 | 0 | 0 | 0 | 0.7 | 0 | 0 | 0 | 0 | 3.4 | 0 | 0.2 | 0.2 | 0 | 0.2 |
| 82 | 0 | 0 | 0 | 0 | 0 | 0 | 0 | 0 | 0 | 0 | 0 | 0 | 0 | 0 |
| 90 | 0 | 0 | 0 | 544.2 | 0 | 0 | 653 | 0 | 217.7 | 435.3 | 108.8 | 108.8 | 0 | 108.8 |
| 100 | 0 | 0 | 0 | 1362.7 | 0 | 0 | 681.4 | 340.7 | 170.3 | 170.3 | 340.7 | 170.3 | 0 | 170.3 |
| 110 | 0 | 0 | 0 | 650.8 | 0 | 0 | 108.5 | 54.2 | 0 | 54.2 | 108.5 | 54.2 | 0 | 54.2 |
| 120 | 2181.1 | 0 | 0 | 2181.1 | 0 | 0 | 0 | 0 | 0 | 0 | 2181.1 | 2181.1 | 0 | 2181.1 |
| 121 | 47.7 | 0 | 0 | 47.7 | 0 | 0 | 0 | 0 | 0 | 0 | 0 | 71.6 | 0 | 71.6 |
| 122 | 433.2 | 0 | 0 | 433.2 | 0 | 0 | 0 | 0 | 0 | 0 | 1299.7 | 0 | 0 | 0 |
| 130 | 0 | 0 | 0 | 13703.6 | 0 | 0 | 0 | 0 | 0 | 0 | 0 | 0 | 0 | 0 |
| 140 | 0 | 0 | 0 | 1491 | 0 | 0 | 0 | 0 | 0 | 0 | 0 | 0 | 0 | 0 |
| 150 | 7180.2 | 0 | 0 | 422.4 | 0 | 0 | 253.4 | 84.5 | 0 | 84.5 | 253.4 | 84.5 | 0 | 84.5 |
| 151 | 0 | 0 | 0 | 0 | 0 | 0 | 0 | 0 | 0 | 0 | 0 | 0 | 0 | 0 |
| 152 | 62.4 | 0 | 0 | 3.7 | 0 | 0 | 0 | 0 | 0 | 0 | 4.4 | 1.5 | 0 | 1.5 |
| 153 | 320.1 | 0 | 0 | 56.5 | 0 | 0 | 0 | 0 | 0 | 0 | 0 | 0 | 0 | 0 |
| 160 | 0 | 0 | 0 | 224 | 0 | 0 | 335.9 | 335.9 | 0 | 0 | 0 | 0 | 0 | 0 |
| 170 | 0 | 0 | 0 | 0 | 0 | 0 | 0 | 164.1 | 0 | 0 | 0 | 54.7 | 0 | 0 |
| 180 | 0 | 0 | 0 | 1150.8 | 0 | 0 | 0 | 0 | 0 | 0 | 479.5 | 0 | 0 | 287.7 |
| 190 | 0 | 647.1 | 0 | 0 | 0 | 0 | 0 | 0 | 0 | 0 | 0 | 0 | 0 | 0 |
| 200 | 19170.2 | 0 | 0 | 0 | 0 | 0 | 0 | 0 | 0 | 0 | 0 | 0 | 0 | 0 |
| 201 | 109.1 | 0 | 0 | 0 | 0 | 0 | 0 | 0 | 0 | 0 | 0 | 0 | 0 | 0 |
| 202 | 97.3 | 0 | 0 | 0 | 0 | 0 | 0 | 0 | 0 | 0 | 0 | 0 | 0 | 0 |
| 210 | 0 | 0 | 0 | 0 | 0 | 366225 | 0 | 0 | 0 | 0 | 0 | 0 | 0 | 0 |
| 220 | 0 | 0 | 0 | 0 | 14694.2 | 0 | 0 | 0 | 0 | 0 | 0 | 0 | 0 | 0 |




[3] For the water body class (code 210), the water PFT area includes 3,110,600 km² of inland water.





**Table A2.** Percentage PFT composition by class based on the most recent update to the global cross-walking table.

| Class code | Bare soil | Built | Managed grasses | Natural grasses | Snow/ice | Water | BD trees | BE trees | ND trees | NE trees | BD shrubs | BE shrubs | ND shrubs | NE shrubs |
|---|---|---|---|---|---|---|---|---|---|---|---|---|---|---|
| 10 | 90 | 0 | 90 | 10 | 0 | 0 | 0 | 0 | 0 | 0 | 0 | 0 | 0 | 0 |
| 11 | 90 | 0 | 90 | 10 | 0 | 0 | 0 | 0 | 0 | 0 | 0 | 0 | 0 | 0 |
| 12 | 30 | 0 | 30 | 0 | 0 | 0 | 0 | 0 | 0 | 0 | 0 | 0 | 0 | 0 |
| 20 | 90 | 0 | 90 | 10 | 0 | 0 | 0 | 0 | 0 | 0 | 0 | 0 | 0 | 0 |
| 30 | 60 | 0 | 60 | 15 | 0 | 0 | 5 | 5 | 0 | 0 | 5 | 5 | 0 | 5 |
| 40 | 20 | 0 | 20 | 30 | 0 | 0 | 7.5 | 7.5 | 0 | 0 | 10 | 10 | 0 | 10 |
| 50 | 0 | 0 | 0 | 0 | 0 | 0 | 0 | 90 | 0 | 0 | 5 | 5 | 0 | 0 |
| 60 | 0 | 0 | 0 | 30 | 0 | 0 | 50 | 0 | 0 | 0 | 0 | 0 | 0 | 0 |
| 61 | 0 | 0 | 0 | 15 | 0 | 0 | 70 | 0 | 0 | 0 | 0 | 0 | 0 | 0 |
| 62 | 0 | 0 | 0 | 45 | 0 | 0 | 30 | 0 | 0 | 0 | 0 | 0 | 0 | 0 |
| 70 | 0 | 0 | 0 | 15 | 0 | 0 | 0 | 0 | 70 | 70 | 5 | 5 | 0 | 5 |
| 71 | 0 | 0 | 0 | 15 | 0 | 0 | 0 | 0 | 70 | 70 | 5 | 5 | 0 | 5 |
| 72 | 0 | 0 | 0 | 45 | 0 | 0 | 0 | 0 | 30 | 30 | 0 | 0 | 0 | 25 |
| 80 | 0 | 0 | 0 | 30 | 0 | 0 | 0 | 0 | 0 | 0 | 2.5 | 2.5 | 12.5 | 2.5 |
| 81 | 0 | 0 | 0 | 15 | 0 | 0 | 0 | 0 | 0 | 0 | 5 | 5 | 0 | 5 |
| 82 | 0 | 0 | 0 | 45 | 0 | 0 | 0 | 0 | 0 | 0 | 0 | 0 | 25 | 0 |
| 90 | 0 | 0 | 0 | 25 | 0 | 0 | 30 | 0 | 20 | 20 | 5 | 5 | 0 | 5 |
| 100 | 0 | 0 | 0 | 40 | 0 | 0 | 20 | 10 | 5 | 5 | 5 | 5 | 0 | 5 |
| 110 | 0 | 0 | 0 | 60 | 0 | 0 | 10 | 5 | 5 | 5 | 5 | 5 | 0 | 5 |
| 120 | 0 | 0 | 0 | 20 | 0 | 0 | 0 | 0 | 0 | 0 | 20 | 20 | 0 | 20 |
| 121 | 0 | 0 | 0 | 20 | 0 | 0 | 0 | 0 | 0 | 0 | 30 | 30 | 0 | 30 |
| 122 | 0 | 0 | 0 | 20 | 0 | 0 | 0 | 0 | 0 | 0 | 0 | 0 | 0 | 0 |
| 130 | 0 | 0 | 0 | 100 | 0 | 0 | 0 | 0 | 0 | 0 | 0 | 0 | 0 | 0 |
| 140 | 0 | 0 | 0 | 100 | 0 | 0 | 0 | 0 | 0 | 0 | 0 | 0 | 0 | 0 |
| 150 | 0 | 0 | 0 | 5 | 0 | 0 | 3 | 1 | 1 | 1 | 1 | 1 | 0 | 1 |
| 151 | 0 | 0 | 0 | 5 | 0 | 0 | 2 | 0 | 6 | 6 | 0 | 0 | 0 | 0 |
| 152 | 0 | 0 | 0 | 5 | 0 | 0 | 0 | 0 | 0 | 0 | 2 | 2 | 0 | 2 |
| 153 | 0 | 0 | 0 | 15 | 0 | 0 | 0 | 0 | 0 | 0 | 0 | 0 | 0 | 0 |
| 160 | 0 | 0 | 0 | 25 | 0 | 0 | 37.5 | 37.5 | 0 | 0 | 0 | 0 | 0 | 0 |
| 170 | 0 | 0 | 0 | 0 | 0 | 0 | 0 | 75 | 0 | 0 | 25 | 25 | 0 | 0 |
| 180 | 0 | 0 | 0 | 60 | 0 | 0 | 0 | 0 | 0 | 0 | 0 | 0 | 0 | 15 |
| 190 | 0 | 100 | 0 | 0 | 0 | 0 | 0 | 0 | 0 | 0 | 0 | 0 | 0 | 0 |
| 200 | 0 | 0 | 0 | 0 | 0 | 0 | 0 | 0 | 0 | 0 | 0 | 0 | 0 | 0 |
| 201 | 0 | 0 | 0 | 0 | 0 | 0 | 0 | 0 | 0 | 0 | 0 | 0 | 0 | 0 |
| 202 | 0 | 0 | 0 | 0 | 0 | 0 | 0 | 0 | 0 | 0 | 0 | 0 | 0 | 0 |
| 210 | 0 | 0 | 0 | 0 | 0 | 100 | 0 | 0 | 0 | 0 | 0 | 0 | 0 | 0 |
| 220 | 0 | 0 | 0 | 0 | 100 | 0 | 0 | 0 | 0 | 0 | 0 | 0 | 0 | 0 |





**Appendix B: complementary information about the modelling results**

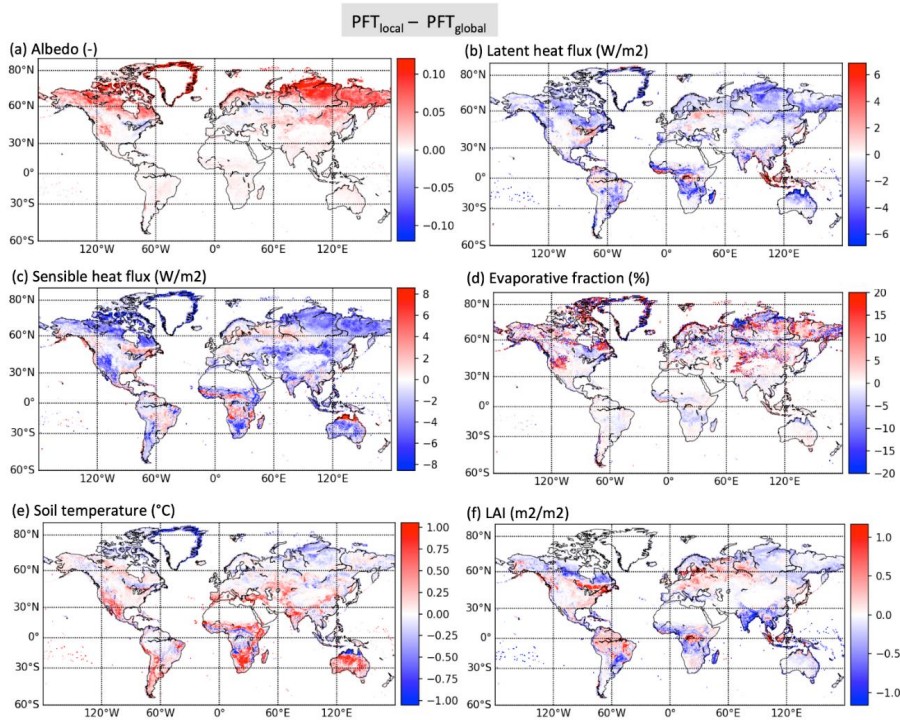


**Figure B1.** Differences in albedo (a), latent heat flux (b), sensible heat flux (c), evaporative fraction (d: Latent heat flux / (Latent + Sensible
heat fluxes), soil surface temperature (e) and Leaf Area Index (LAI, f) simulated by the ORCHIDEE model between the new PFT and the
old PFT distributions, for the annual mean of year 2010 (same as figure 4 but for the annual mean).
**Appendix C: the original default land cover to plant functional type cross-walking table**
**Table C1.** Default land cover to plant functional type cross-walking table provided by the conversion tool with the level 1 UNLCCS
classes and level 2 UNLCCS sub-classes in italics. The units are % coverage of each PFT per UNLCCS class (from Poulter et al., 2015).

| Code | UN LCCS Land Cover Class Description | Trees | | | | Shrubs | | | | Grasses | | Non-vegetated | | |
|------|-------------------------------------|-------|-------|-------|-------|--------|-------|-------|-------|---------|-------|--------------|-------|-----------|
| | | Br Ev | Br De | Ne Ev | Ne De | Br Ev | Br De | Ne Ev | Ne De | Nat Gr | Crops | Bare Soil | Water | Snow/ Ice |
| 10 | Cropland, rainfed | | | | | | | | | | 100 | | | |
| *11* | *Herbaceous cover* | | | | | | | | | | *100* | | | |
| *12* | *Tree or shrub cover* | | | | | | *50* | | | | *50* | | | |
| 20 | Cropland, irrigated or post-flooding | | | | | | | | | | 100 | | | |
| 30 | Mosaic cropland (>50 %) / natural vegetation (tree, shrub, herbaceous cover) (<50 %) | 5 | 5 | | | 5 | 5 | 5 | | 15 | 60 | | | |
| 40 | Mosaic natural vegetation (tree, shrub, herbaceous cover) (>50 %) / cropland (<50 %) | 5 | 5 | | | 7.5 | 10 | 7.5 | | 25 | 40 | | | |
| 50 | Tree cover, broadleaved, evergreen, closed to open (>15 %) | 90 | | | | 5 | 5 | | | | | | | |
| 60 | Tree cover, broadleaved, deciduous, closed to open (>15 %) | | 70 | | | | 15 | | | 15 | | | | |
| *61* | *Tree cover, broadleaved, deciduous, closed (>40 %)* | | *70* | | | | *15* | | | *15* | | | | |




| Code | UN LCCS Land Cover Class Description | Trees | | | | Shrubs | | | | Grasses | | Non-vegetated | | |
|---|---|---|---|---|---|---|---|---|---|---|---|---|---|---|
| | | Br Ev | Br De | Ne Ev | Ne De | Br Ev | Br De | Ne Ev | Ne De | Nat Gr | Crops | Bare Soil | Water | Snow/Ice |
| *62* | *Tree cover, broadleaved, deciduous, open (15-40 %)* | | *30* | | | | *25* | | | *35* | | *10* | | |
| 70 | Tree cover, needleleaved, evergreen, closed to open (>15 %) | | | 70 | | 5 | 5 | 5 | | 15 | | | | |
| *71* | *Tree cover, needleleaved, evergreen, closed (>40 %)* | | | *70* | | *5* | *5* | *5* | | *15* | | | | |
| *72* | *Tree cover, needleleaved, evergreen, open (15-40 %)* | | | *30* | | | *5* | *5* | | *30* | | *30* | | |
| 80 | Tree cover, needleleaved, deciduous, closed to open (>15 %) | | | | 70 | 5 | 5 | 5 | 0 | 15 | | | | |
| *81* | *Tree cover, needleleaved, deciduous, closed (>40 %)* | | | | *70* | *5* | *5* | *5* | | *15* | | | | |
| *82* | *Tree cover, needleleaved, deciduous, open (15-40 %)* | | | | *30* | | *5* | *5* | *0* | *30* | | *30* | | |
| 90 | Tree cover, mixed leaf type (broadleaved and needleleaved) | | 30 | 20 | 10 | 5 | 5 | 5 | | 15 | | 10 | | |
| 100 | Mosaic tree and shrub (>50 %) / herbaceous cover (<50 %) | 10 | 20 | 5 | 5 | 5 | 10 | 5 | | 40 | | | | |
| 110 | Mosaic herbaceous cover (>50 %) / tree and shrub (<50 %) | 5 | 10 | 5 | | 5 | 10 | 5 | | 60 | | | | |
| 120 | Shrubland | | | | | 20 | 20 | 20 | | 20 | | 20 | | |
| *121* | *Shrubland evergreen* | | | | | *30* | | *30* | | *20* | | *20* | | |
| *122* | *Shrubland deciduous* | | | | | | *60* | | | *20* | | *20* | | |
| 130 | Grassland | | | | | | | | | 60 | | 40 | | |
| 140 | Lichens and mosses | | | | | | | | | 60 | | 40 | | |
| 150 | Sparse vegetation (tree, shrub, herbaceous cover) (<15 %) | 1 | 3 | 1 | | 1 | 3 | 1 | | 5 | | 85 | | |
| *151* | *Sparse tree (<15 %)* | | *2* | *6* | *2* | | | | | *5* | | *85* | | |
| *152* | *Sparse shrub (<15 %)* | | | | | *2* | *6* | *2* | | *5* | | *85* | | |
| *153* | *Sparse herbaceous cover (<15 %)* | | | | | | | | | *15* | | *85* | | |
| 160 | Tree cover, flooded, fresh or brakish water | 30 | 30 | | | | | | | 20 | | | 20 | |
| 170 | Tree cover, flooded, saline water | 60 | | | | 20 | | | | | | | 20 | |
| 180 | Shrub or herbaceous cover, flooded, fresh/saline/brakish water | | 5 | 10 | | | 10 | 5 | | 40 | | | 30 | |
| 190 | Urban areas | | 2.5 | 2.5 | | | | | | 15 | | 75 | 5 | |
| 200 | Bare areas | | | | | | | | | | | 100 | | |
| *201* | *Consolidated bare areas* | | | | | | | | | | | *100* | | |
| *202* | *Unconsolidated bare areas* | | | | | | | | | | | *100* | | |
| 210 | Water bodies | | | | | | | | | | | | 100 | |
| 220 | Permanent snow and ice | | | | | | | | | | | | | 100 |
