# Peer review of "A 29-year time series of annual 300-metre resolution plant functional"

_Earth System Science Data, 2022_

## Author Comment (AC1)

**RC1**

**Summary**

This study describes a method used to produce the fractional composition of 14 PFTs at 300m resolution for the ESA-CCI land cover maps over the 1992–2020 period. Several 30m resolution datasets including surface water, tree canopy cover and height are used in the process, which is a significant effort given the large data volume dealt with at the global scale. The authors also compare the use of the new PFT data versus the previous version (based on a generic cross-walking table) in two land surface models for model simulation (ORCHIDEE) and evaluation (JULES), to demonstrate the impact of the new PFT data.

Overall the paper reads well, and the new PFT product is potentially very useful to the climate/land surface modelling community. However, I have a major concern about the accuracy and consistency of the high-res datasets used to derive the fractional composition of the PFTs, especially when the data values (e.g. tree cover and surface water products) are directly used to produce the PFT fractions at 300m pixels. Details are outlined below, which will hopefully improve the future version of the paper.

**Major comments**

(1) In the PFT product, the percentage of tree cover at the 300 m pixel is estimated using the 30 m tree cover data for 2010 from Hansen et al. (2013). Thus, the accuracy of the tree cover in the PFTs is directly linked to the accuracy of the 30 m data from Hansen et al. (2013). **Several previous studies showed that compared to field and other data sources (e.g. Lidar) tree cover data from Hansen et al. (2013) overestimated tree cover in their studied regions.**

Tang et al (2019) showed that in the Sierra national forests USA the tree canopy cover from Hansen et al (2013) overestimated tree cover **with RMSE around 20% when compared to field measurements, and with RMSE nearly 30% when compared to airborne Lidar estimates.** Potapov et al (2015) found that the tree cover product from Hansen et al. (2013) **overestimated tree canopy cover within the peat bog areas in Eastern Europe**. They had to define "forest cover" using a tree canopy cover threshold of >= 49%. Wang et al (2019) also showed that tree cover from Hansen et al (2013) was overestimated in wetland environments over Canada. Though the tree cover data used in Potapov et al (2015) and Wang et al (2019) was for the year 2000, it was produced using the same method as for the 2010 data.

Tang, H., Song, X.-P., Zhao, F. A., Strahler, A. H., Schaaf, C. L., Goetz, S., Huang, C., Hansen, M. C., and Dubayah, R.: Definition and measurement of tree cover: A comparative analysis of field-, lidar- and landsat-based tree cover estimations in the Sierra national forests, USA, Agr. Forest Meteorol., 268, 258–268, https://doi.org/10.1016/j.agrformet.2019.01.024, 2019.

Potapov, P., Turubanova, S., Tyukavina, A., Krylov, A., McCarty, J., Radeloff, V., and Hansen, M., 2015, Eastern Europe's forest cover dynamics from 1985 to 2012 quantified from the full Landsat archive: Remote Sensing of Environment, v. 159, p. 28-43, at http://www.sciencedirect.com/science/article/pii/S0034425714004817.

Wang, L., Bartlett, P., Pouliot, D., Chan, E., Lamarche, C., Wulder, M. A., Defourny, P., & Brady, M. (2019). Comparison and Assessment of Regional and Global Land Cover Datasets for Use in CLASS over Canada. Remote Sensing, 11(19), 2286.

Therefore, these uncertainties in the tree cover data would propagate to the derived PFT product. I wonder whether the Hansen et al (2013) tree cover data and the proposed method are best for producing the fractional composition of PFTs? Two other 30m tree cover datasets (NLCD and GLCF) were included in Tang et al (2019), though they did not perform better than the Hansen et al (2013) data in the evaluation. **I wonder would an ensemble approach using all three of the 30m tree cover datasets as inputs for producing the PFTs be better? Just a thought.**

The question of the quality of auxiliary data and the propagation of errors is important and legitimate. Some papers, including those you have cited in addition to the more recent Potapov et al. (2022), suggest that the Hansen et al. (2013) dataset overestimates the amount of tree cover in some regions. The latter paper (Potapov et al. 2022, see Table 7) indeed confirms that the overall accuracy associated with quantifying forest extent increases from 92.4 to 95.1% when increasing the tree canopy cover threshold from 10 to 30%. Using 30% (rather than 10%) tree canopy cover to define forest reduces commission errors by 5.4%. In our experience, coupled with the fact that we harmonize the derived PFT fractions with the CCI MRLC dataset, an overall accuracy of 92.4%, even with a user accuracy of 80.4%, is a satisfactory degree of accuracy for the Hansen et al. (2013) auxiliary dataset.

To reduce spatial inconsistencies, we only considered global products for use as ancillary inputs. The NLCD product does not satisfy this criterion. The Global Land Cover Facility (GLCF) does generate a global tree cover dataset at 30m resolution; like other datasets, its performance varies according to the tree cover canopy threshold. For example, Feng et al. (2016) show that the overall accuracy of the GLCF dataset drops below 80% for tree cover canopies of 10-35% (Figure 6), with user accuracies dropping drastically with lower tree canopy covers (user accuracy >80% using tree cover canopy >35%; 50% using tree cover canopy of 35%; < 30% using tree cover canopy <35%). Put simply, all products have some weaknesses that must be weighed in the selection of data products.

The purpose of this paper is not to independently validate the input datasets. As explained in more detail in the manuscript text, using the selected auxiliary datasets (all of which are associated with specific accuracies), we calculated an initial PFT distribution at 300 m resolution. To assign tree percentage at 300 m, we used a median aggregation of the contributing 30 m pixels to smooth out the effects of extreme tree canopy cover values. We then undertook a harmonization step so that the final derived PFT percentages in the 300 m pixels were aligned with the CCI MRLC class legend. Importantly, the CCI MRLC dataset has been quantitatively and independently validated (C3S PQAR for LC, full reference given below); the CCI LC validation database is able to validate the land cover classes at 300 m over a range of tree cover values to the precision indicated in the CCI MRLC legend (i.e., based on ranges such as '10 to 30%').

References:
P. Potapov, M. C. Hansen, A. Pickens, A. Hernandez-Serna, A. Tyukavina, S. Turubanova, V. Zalles, X. Li, A. Khan, F. Stolle, N. Harris, X.-P. Song, A. Baggett, I. Kommareddy, A. Kommareddy, The Global 2000-2020 Land Cover and Land Use Change Dataset Derived From the Landsat Archive: First Results. Front. Remote Sens. 3, 1–22 (2022).

C3S PQAR for LC:  Copernicus Climate Datastore the Product Quality Assessment Report ICDR Land Cover 2016-2020. https://datastore.copernicus-climate.eu/documents/satellite-land-cover/D5.2.2_PQAR_ICDR_LC_v2.1.x_PRODUCTS_v1.0.pdf

M. Feng, J. O. Sexton, C. Huang, A. Anand, S. Channan, X. P. Song, D. X. Song, D. H. Kim, P. Noojipady, J. R. Townshend, Earth science data records of global forest cover and change: Assessment of accuracy in 1990, 2000, and 2005 epochs. Remote Sens. Environ. 184, 73–85 (2016).

(2) L576-578, "Since the PFT local product is built mainly for application to land surface models, the actual presence of grass vegetation vs. bare soil will be determined by the model given simulated or prescribed local climate conditions." **This is likely the case when vegetation cover is dynamically simulated (with competition between PFTs)** in the models. However, that is not always the case especially considering that prescribed PFTs are in general more realistic than dynamically simulated ones. For example, the majority of models participating in the TRENDY (trends in net land atmosphere carbon exchanges) **project use prescribed PFTs** without competition between PFTs in their simulations, which contribute to the annual Global Carbon Project's analysis of the land carbon sink (Friedlingstein et al., 2020). The simulations by ORCHIDEE in this paper also use prescribed PFTs. As demonstrated in Fig.3 of this paper and also in Hartley et al (2017), changes in the PFT fractional distribution exert significant impacts on the simulated water, energy and carbon fluxes. **I am not convinced that it's not important to differentiate bare soil from grassland in the PFTs. If the PFT product is intended for use only in model simulations with dynamic competition between PFTs, it needs to be stated explicitly in the paper.**

In ORCHIDEE, the PFT fractions are prescribed annually and do not change throughout the year. The bare soil PFT represents the fraction of the grid cell where vegetation can't grow regardless of the environmental conditions. As is common for LSMs, the seasonal cycle of grasslands and crops is represented in the model. The vegetation cover fraction is calculated daily according to the amount of above ground biomass. Some amount of bare soil can then result from the absence of vegetation in areas that could support vegetation. This fraction of bare soil is presently processed in the same way as the prescribed fraction of pure bare soil (i.e., the bare soil that is unable to support vegetation), but ongoing developments aim to differentiate the treatment of these two fractions since they don't see the same atmospheric conditions (e.g., different radiation resulting from the interactions with surrounding vegetation).

In development of the new PFT product, in shrubland areas, we considered that the amount of bare soil is the result of the seasonality or the dryness of the grasslands or croplands and could be different with higher water availability. This is the reason why we assumed that a shrubland pixel does not include a fraction of bare soil on which vegetation cannot grow. We agree that it is very important to differentiate grasslands from bare soils in land surface models, because they show very different surface properties (albedo, roughness, evaporation capacities, etc.) and will lead to different surface variables and fluxes.

We've updated the manuscript to better describe the bare soil PFT.

Previous version of lines 576-578: "Since the PFT$_{local}$ product is built mainly for application to land surface models, the actual presence of grass vegetation vs. bare soil will be determined by the model given simulated or prescribed local climate conditions."

Changed to: "In the PFT$_{local}$ product, the bare soil PFT represents areas that are not expected to support vegetation regardless of environmental conditions. For shrubland class pixels, we assume that vegetation growth can be supported given the appropriate environmental conditions; therefore, the residual pixel area (after accounting for inland water, tree, and shrub cover) is assigned as natural grass PFT. Since the PFT local product is built mainly for application to land surface models, the actual presence of grass vegetation vs. bare soil for such pixels (of the shrubland class, but also of the other vegetated classes) will be determined by the model given simulated or prescribed local climate conditions. Users should consider the definition of the bare soil PFT to determine suitability of the data product for their use case."

**Minor comments**

Abstract, L16, 2D is not defined previously.

We have replaced "2D" with "two-dimensional" in both instances of its use in the abstract as this is the only sentence in the manuscript where "2D" is used.

L95-96, I'd suggest to modify "this work aims to reduce the cross-walking component of uncertainty" to "this work aims to reduce the uncertainty in the cross-walking component"

Reworded as suggested.

L100, "…with existing high-resolution auxiliary data products that individually characterize one surface type with high accuracy." The authors need to provide the accuracy information of the auxiliary data products in Section 2.1 to support this argument, which are important for users to understand the uncertainties in the PFT product.

We have added the following (L415): There is uncertainty inherent in all datasets, including both the CCI MRLC class data product and the suite of ancillary data products used to derive the pixel-level PFT fractional composition. For example, Potapov et al. 2022 recently analyzed the tree canopy cover product of the Hansen et al. 2013 products according to increasing fractions of canopy cover (above or equal to 10, 20 or 30 %) (see Table 7.). Overall accuracies fall around 90 % irrespective of the selected threshold. User's accuracies and producer accuracies range around 80 % and 88 %  in the worst scenarios (10 % and 30 % respectively). For their surface water product, Pekel et al. 2016 report > 99% accuracy against both omission & commission errors for permanent water, with slightly lower accuracy for seasonal water (>98 % against errors of commission and 73.8-77.4 % against errors of omission, depending on the sensor). Pesaresi et al. 2013 report accuracies > 90 % for their built product, while Potapov et al. 2021 suggest overall accuracies around 88 % for the validation of the Landsat-based forest height map using a height threshold of above or equal to 5 m validated with GEDI RH95 and ALS-based forest height validation data (see Table 2). Clearly, many of the input datasets can claim high degrees of accuracy; yet, any errors in the input datasets may translate to inaccuracies in the PFT data product. Aligning the PFT percentages with the expected fractional cover from the class legend maintains consistency between the PFT product and the CCI MRLC class product.

References

Potapov, P., Li, X., Hernandez-Serna, A., Tyukavina, A., Hansen, M. C., Kommareddy, A., Pickens, A., Turubanova, S., Tang, H., Silva, C. E., Armston, J., Dubayah, R., Blair, J. B., & Hofton, M. (2021). Mapping global forest canopy height through integration of GEDI and Landsat data. Remote Sensing of Environment, 253(October 2020), 112165.

%

L152, "This CCI PFT product is based on v2.0.8 of the CCI MRLC time series", I can only find v2.0.7 data at https://maps.elie.ucl.ac.be/CCI/viewer/. Are v2.0.8 data available to users?

We expect to release v2.0.8 of the CCI MRCL dataset after the (to be submitted) Defourny et al. paper is accepted for publication. Under specific agreements of collaboration, we would consider providing earlier access to the dataset. Such requests should be made to: contact@esa-landcover-cci.org.

L230-235, Table A2 shows that there are small fractions for the shrub PFTs for classes 30-110, which seem to be in contradiction with the description here, i.e. "Pixels belonging to the shrubland classes (codes 120–122 and 180) can have a mixture of trees, shrubs, and herbaceous cover. For pixels of non-shrubland vegetation containing classes, the vegetated portion of the pixel is composed of trees and herbaceous cover". Can you explain?

Table A2 refers to the most recent version of the global cross-walking table for the CCI MRLC (Lurton et al., 2020) where the PFT proportions have been defined on the basis of expert knowledge, taking into account the CCI MRLC legend. The description in L230-235 refers to the decision rules applied to the new PFTlocal dataset. Tables 2 and 3 refer to the new PFTlocal dataset and correctly indicate that the new dataset includes shrub PFT only in shrubland class pixels.

L258-270, can you add the upper and/or lower limits in the text? They are not always included in the legend in Table 1.

We added to the manuscript text the upper & lower limits that were applied for the tree cover classes for the harmonization steps.

Added at end of case #2 (L261): For classes 62, 72, and 82, the legend upper limit is 40%. For classes 50, 60, 61, 70, 71, 80, 81, 90, 160, and 170, the legend upper limit is 100%, and the initial mean tree fraction for the window can never exceed this threshold.

Added at end of case #3 (L263): For classes 50, 60, 62, 70, 72, 80, 82, 90, 160, and 170, the legend lower limit is 16%. For classes 61, 71, and 81, the legend lower limit is 41%.

L298-300, as I understand it, the sparse vegetation classes (150-153) may have some small trees but perhaps more likely to have shrubs than trees, especially if they are located above the tree line, please take a look at the Circumpolar Arctic Vegetation Map https://www.caff.is/flora-cfg/circumpolar-arctic-vegetation-map.

In the CCI land cover classification, the sparse vegetation class 150 (and subclasses 151, 152, and 153) indicates that the 300 m pixel is covered by a maximum of 14% vegetation cover. This vegetation may consist of trees, shrubs, or grasses. In pixels where the distinction of life form is not possible, class 150 is indicated. Classes 151, 152, and 153 correspond, respectively, to sparse tree cover, sparse shrub cover, and sparse herbaceous cover in cases where the life form can reasonably be determined.

In the new PFTlocal product, a non-zero tree percentage is indicated only in cases where the auxiliary Hansen et al. (2013) data product indicates the presence of trees. The tree cover percentage is restricted to <15% to align with the class legend. Because we don't separately quantify the percentage of shrubs, owing to the lack of an appropriate ancillary dataset, we may indeed be underestimating the woody biomass in some sparse vegetation pixels, particularly those of class 152. However, class 152 accounts for less than 1% of the global area of sparse vegetation, and a maximum of 14% vegetation cover (accounting for all vegetation types) is present, by definition, in such pixels.

L357, "The bare area classes (codes 200, 201, and 202) can have up to 3 % vegetation cover, by definition", this vegetation cover information is not shown in Table 1. Can you add such cover information (e.g. 3% etc.) mentioned throughout the paper in Table 1? So that it'd be easier for readers to understand the class codes and the definitions. In addition, I'd suggest to provide a reference.

The land cover class definitions are based on the FAO Land Cover Classification System 2 (LCCS 2). The vegetation fraction used to determine primarily vegetated classes is above or equal to 4 %, following the FAO LCCS 2 (Table 1, "Distinction at the main Dichotomous level and the second level"): "This class applies to areas that have a vegetative cover of at least 4% for at least two months of the year." Reference available online at: https://www.fao.org/3/x0596e/x0596e01f.htm#p310_30093. Therefore, the upper limit for the vegetation fraction in the abiotic "bare class" is 3%. We have added the reference to the FAO LCCS and clarified this aspect at the top of section 2.2.1 (L222): "The vegetation thresholds used to define whether pixels are predominantly vegetated or abiotic are based on the definitions of the CCI MRLC classes, which are based on the concepts and definitions of the FAO LCCS (Di Gregorio and Jansen, 2005)." We have additionally amended the sentence that you have quoted to (L357): "The bare area classes (codes 200, 201, and 202) can have up to 3 % vegetation cover (by definition of the abiotic class in the FAO LCCS, Di Gregorio and Jansen, 2005), so bare area pixels can have non-zero fractions of bare soil, tree, and water PFTs." We have also made note of the vegetation threshold for the bare class in Table 1; the Class description column for Class 200 now reads: "Bare areas (total vegetative cover < 4%)."

L360-361, "the latter of which is estimated as 100 % minus the inland water percentage", this seems to be too high since the productivity of mosses and lichens is in general much lower than grasses. I'd suggest the authors to consult a LSM expert on this.

Since the ground in such pixels is typically densely covered by lichens and mosses, we find it appropriate to estimate the fraction of lichens & mosses as 100% minus the inland water percentage. Since lichens & mosses are not one of our 14 PFTs, owing to this being a less common PFT among ESMs and LSMs, we have assigned this vegetation type to the grass PFT given the low biomass of both lichens & mosses and grasses. Based on the atmospheric conditions in this region, an LSM will assign a low LAI and low productivity to these pixels. Furthermore, the CCI MRLC dataset includes a class specifically for lichens & mosses (class 140); therefore, individual users can determine if an adjustment to pixels of this class is needed given their specific LSM framework.

L379, 2° × 2° is rather large, I wonder how many pixels are determined this way? I'd suggest to provide a percentage.

Good suggestion. Only a tiny number of pixels have had the tree type (that is, broadleaved evergreen, broadleaved deciduous, needleleaved evergreen, or needleleaved deciduous) assigned using a window of 1° x 1°or larger. The following table indicates the percentages of

pixels that had tree type defined using windows of various sizes (or directly using the class legend, meaning no window calculation was needed). We added these percentages at the end of the paragraph (L380): "The vast majority (75%) of the pixels with a non-zero tree fraction were assigned a tree type directly using the class legend; an additional 24% had tree type assigned using a surrounding window of 0.25° × 0.25° , < 1 % using a larger window up to a size of 1° × 1°, and < 0.1% using an even larger window up to a size of 2° × 2°."

Table: Percentages of pixels according to the source of information for the tree PFT assignment (BE, BD, NE, or ND). The source can be the pixel legend or the legend of adjacent pixels observed in various window size expansions. Only relevant for pixels that have never changed class.

| Source of tree PFT assignment | Fraction [%] |
|---|---|
| pixel class legend (no window calculation necessary) | 74.68 |
| window of 0.25° lon x 0.25° lat | 24.33 |
| window of 0.5° lon x 0.5° lat | 0.61 |
| window of 0.75° lon x 0.75° lat | 0.20 |
| window of 1.0° lon x 1.0° lat | 0.09 |
| window of 1.25° lon x 1.25° lat | 0.05 |
| window of 1.5° lon x 1.5° lat | 0.03 |
| window of 1.75° lon x 1.75° lat | 0.02 |
| window of 2.0° lon x 2.0° lat | 0.00 |

L410-411, "5) 96 % bare soil PFT and 4 % natural grass PFT (to meet the legend minimum of vegetation cover) are assigned to pixels of the sparse vegetation classes", should this be bare classes? Though previously described as "can have up to 3 % vegetation cover" instead of 4%.

This section describes how PFT fractions are assigned for the pixels that exist outside of the extents of the input auxiliary data products. There are very few pixels in this zone that belong to a class other than water or snow/ice. In the case that such a pixel belongs to the bare soil class, then we assume 100% bare soil PFT (as denoted in our case # 3). As described in a comment further above, note that bare class pixels can have up to 3% vegetation cover; here, we assume 0% vegetation cover. In the case that a pixel in this zone belongs to the sparse vegetation class, then we assume 4% vegetation cover (as in our case #5). As described above, sparse vegetation pixels can have 4-14% vegetation cover; here, we assume the minimum of this range (4%). The minimum threshold of 4% vegetation for the sparse vegetation class (and 3% maximum for the bare class) corresponds to the FAO LCCS 2 definition (Di Gregorio and Jansen, 2005).

Di Gregorio, A., and Jansen, L. J. M. Land Cover Classification System (LCCS): Classification Concepts and User Manual. Fao (Vol. 53). Food & Agriculture Organization, 2005.

Fig.1 (c), some needleleaved evergreen trees are distributed above the treeline, is this realistic? Are there field data or references to support this?

The question of the tree line is interesting and worth monitoring as it is likely to move northward as a consequence of climate warming. The PFT product does indeed indicate some needleleaved evergreen trees north of the polygons segmented in the Circumpolar Arctic Vegetation Map. In such cases, the non-zero tree fractions generally occur in pixels that are classified as a needleleaved evergreen tree cover class in the CCI MRLC classification, and the tree canopy cover in the PFT dataset is generally quite low (around 16%, which is the minimum tree density allowed for such pixels by class definition). It is likely that, in the MRLC class product, some shrubs in this region are misclassified as trees given the spatial scale of the product, combined with the lack of solar illumination and the lack of cloud-free and snow-free observations over these high latitudes. Indeed, comparison against the 20-m GlobPermafrost map (Bartsch et al., 2019) based on Sentinel data suggests that the needleleaved evergreen tree cover class in the CCI land cover product may be overestimated in the "Shrub, Tundra" area. Your comment draws the attention of the CCI MLRC team to improve the tree representation in the high latitudes. Nonetheless, the PFT product remains consistent with the CCI MRLC classification. We have added a sentence in the manuscript to reflect the potential overestimation of the needleleaved evergreen tree cover class in this region (line 511): "Because the PFT product is harmonized with the CCI MRLC class product, potential classification errors can impact the PFT product. For example, recent high-resolution mapping in the circumpolar Arctic (Bartsch et al. 2019) suggests that the CCI MRLC classification may overestimate needleleaved evergreen tree cover in this region, resulting in a possible overestimate of the tree PFT percentage in such pixels. Future improvements to the land cover classification will likewise flow through to the PFT product." We additionally added a final sentence to the manuscript (L763): "Because the PFT product is harmonized with the CCI MRLC map series, future improvements in the land cover product will flow through to the PFT product."

Bartsch, Annett; Widhalm, Barbara; Pointner, Georg; Ermokhina, Ksenia A; Leibman, Marina; Heim, Birgit (2019): Landcover derived from Sentinel-1 and Sentinel-2 satellite data (2015-2018) for subarctic and arctic environments. Zentralanstalt für Meteorologie und Geodynamik, Wien, PANGAEA, https://doi.org/10.1594/PANGAEA.897916

Fig.1(d) seems to show more coverage for Needleleaved deciduous trees than in the CCI Viewer and the tree cover map in Hansen et al (2013), can you explain why?

We confirmed that non-zero needleleaved deciduous tree cover occurs only for the CCI MRLC class 80 at 300 m resolution. Any visual impression of enhanced coverage of pixels containing needleleaved deciduous tree PFT is likely related to the fact that Figure 1 is based on a 0.25° x 0.25° window grid.

Fig. 1(g) and (h), there are large extent of needleleaved evergreen/deciduous shrubs, are there field data or literature to support this? I am not aware of the use of these PFTs in any models.

We aim to provide a dataset that is maximally useful to as many modelling teams as possible. Important differentiators of PFTs include life form (tree, shrub, grass), leaf type (needleleaved, broadleaved), and phenology (evergreen, deciduous). Thus, dividing the shrubs into such categories is a natural framework for the PFT dataset. While LSMs may not currently use this division for shrubs, they may do so in the future as they are continually being updated. Furthermore, as shrubs are handled differently by different models today, providing this disaggregation provides a flexible dataset that can be applied in various ways

to the different model frameworks. For example, teams can collapse the shrub categories by phenology or leaf type to match their model representation, or they can combine the fully disaggregated categories with the corresponding tree types in models where shrubs are not represented separately from trees.

I'd suggest to use a scale bar with more levels, and perhaps the same scale bar can be used for the different PFTs maps in Fig.1.

The spatial illustration of the PFTs was rather tricky. All figures share the same continuous scale from 0 to 100%, except for the PFT of permanent snow and ice cover which has only two values (0 or 100% cover). The continuous scale offers the most levels. We tried many different visualization variants but found that the current option best highlighted the spatial details of each of the PFTs at the global scale.

L538, "grass vegetation may be assigned in some cases that might otherwise be a temporary bare area", can you elaborate a bit on this? How do you know that it might be "temporary bare area" vs. permanent bare area?

For vegetated surfaces, there could be particular events that prevent vegetation from growing from one year to the next or for all months of the year. For example, areas of natural and managed grasses could be temporary bare areas for a particular year if the weather conditions of a specific year did not allow for vegetation to develop to at least 4% of vegetation (e.g., a significant period of drought). This could also be the case for managed grasses if, in a given year, the cultivated land was sown but failed to develop or if the field was prepared but not sown.

Section 4.2, it seems to me that there is not enough evidence to show that the new PFTs are more realistic than the previous ones. Thus it is hard to interpret results shown in Fig.5.

In section 4.2, we suggest that the PFT product can serve as a benchmarking dataset for models given the overall reduction in bias between the JULES-TRIFFID results and the new PFT product (relative to the bias between JULES-TRIFFID and the original cross-walking table). The bias reduction is especially strong for the shrub cover type.

The reduction in bias isn't surprising - the original CWT assigns the same PFT fractional composition to all pixels within a class, removing intraclass spatial variability in PFT composition. In contrast, the new PFT product exhibits spatial variability within classes, using published high-resolution datasets to guide the assignment of fractional composition.

Table A2, note sum of fractions are either greater than 100% or <100% for some classes (e.g.10-40).

Thank you for spotting these errors! We have corrected the table.

---

## Author Comment (AC2)

**RC2**

**General comments**

The article is composed overall well and makes a useful support for the publication of the dataset. The method of using higher resolution, specialized data sets is appropriately chosen to refine the sometimes very vague class definitions of the ESA CCI land cover time series. The additional extension of the CCI user tool in order to enable the user to translate the CCI land cover classes to individual PFT maps addresses the needs of the regional climate model community, where different model families have different requirements to the land cover input.

The significance of such a dataset is paramount for the climate modelling community. The integration of the information of multiple high-resolution, remotely sensed datasets into the well-known ESA-CCI land cover time series certainly increases the potential high quality of the PFT time series. **However, all additional input datasets as well as the baseline ESA CCI incorporate uncertainties which are partially mentioned in the original dataset publications or investigated and published by the user community and should be at least mentioned in the present work. Therefore, I would suggest focusing section 3 more on the dataset accuracy aspect then on the comparison to the original PFT$_{global}$ distribution.**

It is found that the cross-walking uncertainty is higher than the land cover product uncertainty itself (Hartley et al. 2017). Yet what is missing is an investigation of the quality of the final product. In addition to the use of the newly developed PFT dataset into RCM experiments and the comparison to the original ESA PFT cross-walking results, a validation through comparison to independent data should be an essential part of this effort. For example, within the GLOBCOVER initiative, the product was compared to a dedicated reference database (Defourny et al. 2009).

Note that a quantitative validation of the CCI medium-resolution land cover class dataset is available (C3S PQAR for LC, full reference given below). In building the PFT dataset from ancillary datasets, we take advantage of the high quantified accuracy of the land cover dataset and align the PFT fractional covers with the expectations for each class according to the class legends. Below, we provide some additional flavour regarding how well the ancillary datasets align with the class legend & the frequency with which adjustment to the PFT percentage was necessary to achieve alignment. We select the tree cover classes for this analysis.

Using all 300 m pixels of the CCI MRLC dataset that fall within the extent covered by the 30 m Hansen et al. 2013 tree cover dataset, we compared (1) the initial tree cover percentage at 300 m estimated from the ancillary Hansen et al. dataset (that is, before applying the harmonization procedure that aligns the tree cover fraction with the CCI MRLC class legend) with (2) the expected tree cover percentage based on the class legend. We additionally calculated the mean tree cover percentage across all pixels of a class, based on the ancillary product. We performed this analysis by class for each of the tree cover classes & subclasses. Results are shown in the table below.

Considering the mean tree cover percentage for all pixels within a class, based on the ancillary Hansen et al. 2013 product, only class 82 has a mean tree cover (3%) that falls outside of what is expected based on the class legend (in this case, 15-40%); however, this class has less than two dozen pixels globally.

Class 50 has the largest number of pixels among any of the tree cover classes and exhibits especially strong correspondence between the class legend and the ancillary data product, with 97% of pixels of this class having a calculated tree cover percentage that meets the class legend expectations. The ancillary data product suggests a high mean tree cover percentage of 89% for pixels of this class. (Interestingly, this is quite close to the 90% tree cover suggested by the original global cross-walking table.)

Classes 60, 61, 70, 71, and 81 each have agreement between the legend and ancillary product for at least 80% of pixels. Classes 72 and 82 have low levels of agreement (9% and 10%, respectively), but each of these classes has only a very small number of pixels. Both are subclasses with an expectation of 15-40% tree cover, with the majority of pixels that exhibit a mis-match having a tree cover percentage from the ancillary product that is lower than expected by the legend; in such cases, the alignment process increases the percentage tree cover so that it falls within the range suggested by the legend (see manuscript for method). Class 62 likewise has a legend expectation of 15-40%, but the mis-matched pixels show a more even split between over- and underestimation from the ancillary product.

| Class code | Class description | % of tree cover class pixels belonging to this class | Mean tree cover percent across all pixels of this class, based on ancillary product | % of pixels having tree cover estimated from ancillary product matching legend |
|---|---|---|---|---|
| 50 | Tree cover, BE >15% | 23.1 | 89 | 97 |
| 60 | Tree cover, BD >15% | 13.7 | 61 | 86 |
| 61 | Tree cover, BD >40% | 1.9 | 58 | 80 |
| 62 | Tree cover, BD 15-40% | 6.6 | 29 | 51 |
| 70 | Tree cover, NE >15% | 19.0 | 59 | 85 |
| 71 | Tree cover, NE >40% | 7.8 | 65 | 87 |
| 72 | Tree cover, NE 15-40% | <0.01 | 22 | 9 |
| 80 | Tree cover, ND >15% | 19.2 | 33 | 62 |

| 81 | Tree cover, ND >40% | <0.01 | 73 | 83 |
|---|---|---|---|---|
| 82 | Tree cover, ND 15-40% | 0 | 3 | 10 |
| 90 | Tree cover, mixed tree type | 6.4 | 79 | NA |
| 160 | Tree cover, flooded - fresh or brackish | 1.9 | 70 | NA |
| 170 | Tree cover, flooded - saline | 0.4 | 52 | NA |

A full comparison of the PFT maps with external data is beyond the scope of this paper, but would be a worthy topic for a follow up paper.

C3S PQAR for LC: Copernicus Climate Datastore the Product Quality Assessment Report ICDR Land Cover 2016-2020. https://datastore.copernicus-climate.eu/documents/satellite-land-cover/D5.2.2_PQAR_ICDR_LC_v2.1.x_PRODUCTS_v1.0.pdf

The article presents the workflow with all necessary detail for the user community, which makes the article quite extensive. **For a better overview a graphic outline of the general workflow would be highly beneficial for the reader.**

We have added a new table (below) to the manuscript that serves as a diagram of the method. We introduce the table at L222: "Table 2 is a high-level overview of the method used to derive the PFT fractional composition for the static pixels."

Table 2. Summary of method applied to derive pixel-level functional type composition by land cover class. See Table 1 for more comprehensive class descriptions. PEA16 = surface water data product of Pekel et al. 2016. HEA13 = tree canopy cover product of Hansen et al. 2013. PEA13 = Global Human Settlement Layer from Pesaresi et al. 2013. PEA21 = tree canopy height dataset of Potapov et al. 2021. For the calculation of tree percentage: "Method 1" indicates that, in cases of disagreement in tree cover percentage between the ancillary dataset and the class legend, a window of up to 0.5° x 0.5° is used to estimate the final tree cover percentage based on neighbourhood pixels of the same class; and "Method 3" indicates that an upper limit of 14 % tree cover is applied based on the class definition. See the text for additional details about the processing and use of the ancillary data products, the method used to align the derived PFT percentages with the class legend, the scaling method applied in cases where the sum of PFT percentages from the ancillary data exceeds 100% in a pixel, and the method used to derive the PFT fractional composition for pixels falling outside of the extents of the ancillary datasets.

| Class description | Inland water % | Tree % | Tree type | Grass % | Grass type | Shrub % | Bare soil % | Built % | Snow/ice % |
|---|---|---|---|---|---|---|---|---|---|
| Rainfed cropland (10-12) | PEA16 | HEA13 | Neighbourhood majority | 100% - water % - tree % | Managed | 0% | 0% | 0% | 0% |
| Irrigated or post-flooding cropland (20) | | | | | | | | | |
| Mosaic of cropland and natural vegetation (30) | | | | | | | | | |
| Mosaic of cropland and natural vegetation (40) | | | | | Managed & natural mixture | | | | |
| Mosaic of tree/shrub and herbaceous (100 & 110) | | | | | Natural | | | | |
| Grassland (130) | | | | | | | | | |
| Broadleaved evergreen tree cover (50) | | HEA13, Method 1 | Class legend | | | | | | |
| Broadleaved deciduous tree cover (60-62) | | | | | | | | | |
| Needleleaved evergreen tree cover (70-72) | | | | | | | | | |
| Needleleaved deciduous tree cover (80-82) | | | | | | | | | |

| Class | | | | | | | | |
|---|---|---|---|---|---|---|---|---|
| Mixed leaf type tree cover (90) | | | Neighbourhood majority | | | | | |
| Flooded tree cover (160-170) | | | | | | | | |
| Lichens and mosses (140) | | 0% | N/A | 100% - water % | | | | |
| Sparse vegetation (150-153) | | HEA13, Method 2 | Neighbourhood majority | Tree % + grass % must be in range 4-14% | | | 100% - water % - tree % - grass % | |
| Shrubland (120-122) | | PEA21 | Biogeographical approach | 100% - water % - tree % - shrub % | | PEA21 | 0% | |
| Flooded shrub or herbaceous cover (180) | | | | | | | | |
| Urban areas (190) | | HEA13 | Neighbourhood majority | 100% - water % - tree % - built % | | 0% | | PEA16 |
| Bare areas (200-202) | | | | 0% | N/A | | 100% - water % - tree % | 0% |
| Inland water bodies (210) | | | | 100% - water % - | Natural | | 0% | |

| | | | | tree % | | | | | |
|---|---|---|---|---|---|---|---|---|---|
| Ocean (210) | 100% | 0% | N/A | 0% | N/A | | | | |
| Permanent snow and ice (220) | 0% | | | | | | | | 100% |

**Specific comments**

L197 Sections 2.1.7 and 2.1.8 are missing, please adjust section numbering

Thank you for spotting this. Section numbering was adjusted.

L250f (also L375f) please explain a bit the size of the 0.25° neighborhood window, would a rather smaller window not be more appropriate to the ~300m (and finer) dataset resolution? Did you test smaller sizes?

We selected a window size of 0.25° as an appropriate size for picking up average features of the land cover. We wanted to avoid using a window that was too small to avoid propagating non-representative features of the landscape.